# External Stimuli-Induced Welding of Dynamic Cross-Linked Polymer Networks

**DOI:** 10.3390/polym16050621

**Published:** 2024-02-24

**Authors:** Yun Liu, Sheng Wang, Jidong Dong, Pengfei Huo, Dawei Zhang, Shuaiyuan Han, Jie Yang, Zaixing Jiang

**Affiliations:** 1School of Chemistry and Chemical Engineering, Harbin Institute of Technology, Harbin 150040, China; liuyun99@stu.hit.edu.cn; 2Key Laboratory of Bio-Based Materials Science & Technology of Ministry of Education, Northeast Forestry University, Harbin 150040, China; 13156937590@163.com (S.W.); dongjidong@nefu.edu.cn (J.D.); huopengfei@nefu.edu.cn (P.H.); 3Jiangsu Co-Innovation Center of Efficient Processing and Utilization of Forest Resources, College of Science, Nanjing Forestry University, Nanjing 210037, China

**Keywords:** polymer welding, dynamic cross-linked networks (DCNs), dynamic covalent bonds (DCBs), covalent adaptable networks (CANs), stimuli-responsive

## Abstract

Thermosets have been crucial in modern engineering for decades, finding applications in various industries. Welding cross-linked components are essential in the processing of thermosets for repairing damaged areas or fabricating complex structures. However, the inherent insolubility and infusibility of thermoset materials, attributed to their three-dimensional network structure, pose challenges to welding development. Incorporating dynamic chemical bonds into highly cross-linked networks bridges the gap between thermosets and thermoplastics presenting a promising avenue for innovative welding techniques. External stimuli, including thermal, light, solvent, pH, electric, and magnetic fields, induce dynamic bonds’ breakage and reformation, rendering the cross-linked network malleable. This plasticity facilitates the seamless linkage of two parts to an integral whole, attracting significant attention for potential applications in soft actuators, smart devices, solid batteries, and more. This review provides a comprehensive overview of dynamic bonds employed in welding dynamic cross-linked networks (DCNs). It extensively discusses the classification and fabrication of common epoxy DCNs and acrylate DCNs. Notably, recent advancements in welding processes based on DCNs under external stimuli are detailed, focusing on the welding dynamics among covalent adaptable networks (CANs).

## 1. Introduction

Joining technology is paramount in polymer processing, driven by the intricate and precise structural requirements for extending service life [1,2]. The primary methods of joining encompass mechanical fastening, adhesive, and welding. While traditional mechanical fasteners risk damaging bulk materials, adhesives exhibit weak solvent resistance. In contrast, welding serves as a joining strategy, seamlessly integrating two or more separated materials without screws or adhesives. Over the past few decades, welding processes predominantly have relied on the diffusion and entanglement of molecular chains induced by external energy sources. For instance, Yussuf et al. employed the fusion bonding method to assemble thermoplastic PMMA parts [3], and Lee et al. [4] demonstrated the welding of partially cross-linked PEO networks with a suitable solvent. However, highly cross-linked thermosetting polymers, characterized by three-dimensional networks, impose challenges due to restricted polymer chain mobility, complicating the attainment of satisfactory welding results through traditional approaches. Despite these challenges, their exceptional mechanical properties and resistance have led to diverse applications in engineering. Consequently, the imperative to develop welding technology for cross-linked network polymers has prompted researchers to explore chemical welding strategies driven by chemical bonding.

Introducing dynamic chemical bonds into thermosetting polymers facilitates the cleavage and recombination of bonds under external stimuli, thereby conferring plasticity and weldability to the dynamic cross-linked network (DCNs) [5,6]. Dynamic chemical bonds encompass non-covalent interactions and dynamic covalent bonds (DCBs). Non-covalent interactions exhibit rapid and sensitive reconstruction but are also susceptible to environmental influence and display lower mechanical properties, such as hydrogen bonds [7], coordination bonds [8], and ionic bonds [9]. In contrast, DCBs possess higher bond energy and more excellent stability, garnering increased attention, including dynamic ester bonds [5], dynamic imine bonds [10], dynamic carbamate bonds [11,12], silico–oxygen bonds [13,14,15], Diels–Alder bonds [16], etc. The cross-linked networks incorporating these introduced DCBs are defined as covalent adaptable networks (CANs) [17,18,19,20], representing the most extensively researched DCNs. The welding process and weldability of DCNs within distinct dynamic bonds have attracted increasing attention in recent years.

Over the past two decades, numerous weldable DCNs have been documented, showcasing chemical bonding at the contact interfaces. For instance, Mathieu Capelot et al. [21] engineered weldable epoxy networks based on transesterification and discussed the welding performance under distinct welding conditions. Zhu et al. [22] achieved welding in various acrylate cross-linked networks through dynamic hydrogen and ionic bonding. In addition to these networks, materials such as polyurethane (PU) [12,23,24,25], polybutadiene (PB) [26,27], polyester [28,29], and polyimine (PI) [30,31,32,33], have demonstrated the capability for chemical welding by incorporating dynamic chemical bonds. The weldability of DCNs depends on the chemical structure of the network (e.g., the type of dynamic interaction and the cross-linked density of the network) and processing conditions (e.g., welding temperature and welding time). This diversification opens a novel avenue for the design of weldable engineering materials. Furthermore, DCN composites involving additions of other oligomers [15,34,35] and functional particles [36,37,38] hold promise for enhancing properties and extending stimulus-response capabilities beyond those exhibited by the original networks.

External stimuli commonly engage cross-linked networks, including temperature, light, solvent, pH, electricity, and magnetic fields. DCNs exhibiting diverse structures and components display distinct responses to different stimuli. Taking CANs as exemplars, most CANs demonstrate responsiveness to thermal stimuli by breaking and reforming dynamic chemical bonds at specific temperatures. Light stimuli apply to specific dynamic disulfide networks and composites incorporating fillers with photothermal effects. Solvent-assisted welding is pertinent to CANs whose rearrangement mechanisms involve dynamic imine reactions and transesterification. Under suitable electrical and magnetic stimuli, CAN composites featuring responsive particles can realize the welding process. Additionally, the welding performance under different stimuli is highly contingent on different welding process conditions. By integrating multi-stimulus responsiveness, the assembly of more dissimilar cross-linked networks becomes feasible, thereby fostering more practical applications across various fields.

The chemical welding of cross-linked networks finds myriad applications in diverse fields, including but not limited to wearable sensors [39,40], soft actuators [41], and batteries [42,43,44]. This welding capability empowers cross-linked network polymers to achieve functionalities such as shape-shifting [45], heterogeneous material welding [15], complicated structure assembly [22], and modular 4D printing [46]. Wu et al. [11] integrated multi-material actuators based on dynamic carbamate bonds which demonstrated the promising application prospect in new-generation smart devices. Enormous potential is anticipated in future applications across various domains, including architecture, aerospace, healthcare, machinery, and robotics. Beyond weldability, the dynamic characteristics of cross-linked networks also confer additional processing properties, such as recyclability [47], mallability [33], repairability [48], and degradability [49]. These attributes bridge the gap between thermoplastic and thermoset resins, expanding the repertoire of processing methodologies for three-dimensional networks.

This review is dedicated to examining the distinctive network design, enhanced welding processing, and varied applications of weldable cross-linked networks responsive to external stimuli. We commence by delineating the utilization of dynamic bonds in the design of DCNs, encompassing both non-covalent interactions and reversible covalent bonds. Furthermore, we delve into synthesizing commonly used thermal-polymerization epoxy DCNs and photo-polymerization acrylate DCNs realized through welding techniques. Subsequently, we meticulously scrutinize the processing condition and potential applications of weldable DCNs under external stimuli, focusing on CANs. Finally, we provide a comprehensive summary and future outlook on the topic, encompassing achievements, identified limitations, and prospective developments.

## 2. Dynamic Bonds Used for Welding

The chemical welding of cross-linked networks depends on dynamic bonding among neighboring molecules at the contacting surfaces under certain conditions. The dynamic bond resulting in cross-linked network welding can be either supramolecular interaction or DCBs. The primary attention is placed on the DCBs in this review.

### 2.1. Supramolecular Interactions

Supramolecular interactions realize self-assembly through highly directional and reversible non-covalent bonds, which makes two fully contact networks bond at the interface. However, the overall mechanical property of a material is commonly low. Supramolecular interactions used for chemical welding include hydrogen [50,51], metal–ligand coordination [8,52], host–guest interaction [53,54,55,56], and so on. These reversible interactions may occur spontaneously at an ambient temperature or under external stimuli, which mainly depend on the types of interactions and the structural characteristics of cross-linked networks.

Hydrogen bonding is the most used supramolecular interaction, which binds characteristics ranging from highly dynamic (association constants K_a_ < 100 M^−1^) to quasi-covalent (K_a_ > 10^6^ M^−1^). Highly dynamic characteristics result in bonding at room temperature while possessing poor environmental resistance and mechanical properties [57]. In contrast, quasi-covalent characteristics show better mechanical stability, in which bond cleavage and reformation occur under robust external energy simulation [58]. Furthermore, variable welding studies have also been reported based on hydrogen hybrid interaction with other supramolecular interactions [22,39] or DCBs [59,60,61].

Other supramolecular interactions have also been individually applied for welding cross-linked networks. The self-welding process based on metal–ligand coordination derives from the reversible and oriental reaction between the organic or polymeric ligand and the free metal ions. Furthermore, unlike hydrogen bonding, metal–ligand coordination could respond to multiple stimuli, such as photo [8,62] and solvent [52]. In addition, host–guest interaction is also a powerful approach to achieving welding, which holds great potential for developing medicinal applications due to biocompatibility. Of course, doping some functional particles in the non-covalent system also enables supramolecular network composites to achieve welding under specific stimuli [53].

Supramolecular interactions prefer achieving chemical welding under relatively mild conditions due to their weak bonding energies than DCBs. In other words, the stability of supramolecular interactions can be easily affected, especially in solution, which is difficult to characterize further and investigate. Although their highly dynamic characteristics make them suitable for modular assembly design, many intrinsic mechanisms, mainly related to the time scale of welding kinetics, still need to be determined in-depth [22,58,63,64].

### 2.2. Dynamic Covalent Bonds

Similar to supramolecular interactions, DCBs also show reversible bond formations. Given that the bonding energy of a covalent bond is generally higher than that of a non-covalent bond, the breaking and formation are more difficult, and the trigger conditions are more demanding, which also means that the dynamic covalent system can maintain the stability of the network structure without stimuli. Moreover, the equilibration processes are much slower in dynamic covalent systems than in supramolecular ones [65]. Accordingly, introducing a proper catalyst can promote the system to a thermodynamically stable product on a suitable time scale. Several chemical welding research based on DCBs have been reported through the catalytic action and external stimuli in recent years.

In 2010, Bowman defined covalent adaptable networks (CANs) as cross-linked networks with an abundant number and topology of reversible covalent bonds to make the network system respond chemically to applied stimuli [66]. According to responding bond rearrangement mechanisms, CANs can be divided into two types [67,68]: dissociative and associative mechanisms. For dissociative CANs, a new bond forms behind the old bond breakage. In forming a new topological structure, the cross-linking density decreases first and then increases, rapidly dropping macroscopical viscosity. This type undergoes reversible addition reactions or reversible condensation reactions (Figure 1), and typical reactions include Diels–Alder reaction [16,69,70,71,72], carbene dimerization [73], diarylbibenzofuranone [74], transalkylation [75], etc.

The most researched reaction with a dissociative mechanism is the Diels–Alder reaction (DA reaction), a thermal-reversible [4+2] cycloaddition between a diene and a dienophile. Specifically, the DA reaction between furan and maleimide couple is the most extensively studied [76]. Cyclohexene product can be obtained at about 60 °C, while the retro-DA reaction occurs significantly when the temperature rises above 110 °C. Due to the outstanding selectivity and efficiency of the “click” characteristic, the DA reaction was widely introduced into the thermoset network to perform dynamic characteristics [16,77].

Unlike the dissociative mechanism, a new bond forms with the old bond cleavage for associative CANs (Figure 1). Thus, cross-linking density and network viscosity remain constant during the network rearrangement. The rearrangement of associative CANs undergoes reversible exchange reactions (RERs), which include transesterification of esters [5,28,78,79,80,81,82,83,84] and boronic esters [85], imine exchange [30,86,87,88], transamination [33], transcarbamoylation [25,89], thiol–disulfide exchange [90], olefin metathesis [91,92], silyl ether exchange [93], etc. The kinetics of bond exchange reaction is dependent on temperature and catalyst. Typical dissociative and associative schemes are listed in Table 1.

Transesterification is the most representative class among the multiple types of reversible exchange reactions. In 2011, Leibler and his group [5] first introduced zinc acetate catalyst into the classical epoxy system, which showed cross-linked networks with unprecedented re-processability via topology rearrangement by transesterification. The hard epoxy network was synthesized by a reaction of DGEBA using glutaric anhydride with epoxy/acyl 1:1 in the presence of Zn(acac)_2_. At an ambient temperature, the system behaved like a classical hard epoxy resin. However, unlike traditional epoxy-anhydride resins, there was a transition temperature from a viscoelastic solid to a viscoelastic liquid called topology freezing transition temperature (T_v_). Above T_v_, RERs could be activated only by adding the proper catalyst resulting in topology rearrangements. It was possible to reprocess without precise temperature control, which could be termed a strong glass-former. Therefore, Leibler defined these covalent networks that changed their topologies through thermoactivated bond exchange reactions as “vitrimers” [94].

The rheological behavior of vitrimer is closely associated with the chemical welding processing through RERs. There are two crucial transition temperatures in vitrimer: glass transition temperature (T_g_) and mentioned T_v_. Like amorphous polymers, the network behaves in a glassy state at temperatures below T_g_. At this moment, neither the molecular chain nor the chain segment can move; only the atoms or groups constituting the molecule vibrate at their equilibrium position. With the increasing temperature above T_g_, the motility of the molecular chains is enhanced. As for T_v_, it is the critical temperature at which the reversible transition between “flow” and “freeze” proceeds. Furthermore, the value could be regulated by the RER catalyst type and concentration. In addition, the two mentioned temperatures change when the cross-linking density varies.

Note that compared with CANs, “vitrimers” have certain limitations. For vitrimers, topology rearrangement occurs under thermal simulation (including photothermal effect, magnetothermal effect, Joule heating effect, etc.), which triggered temperature needs to be higher than T_v_. In some literature, vitrimers are also called associative CANs [76,95]. Many dissociative CANs whose viscosity and thermal stability had linear Arrhenius-like behavior in specific temperature ranges have recently been reported [95,96]. These dissociative CANs, called vitrimer-like materials, exhibit similar properties to vitrimers.

By the way, spontaneous interactions between DCBs in the absence of external stimuli have also been reported. Chandan et al. [45] developed a set of self-weldable CANs based on a room-temperature exchangeable thia-Michael adduct. Wang et al. [97] designed recyclable polythiourethane which could achieve complete self-healing at room temperature after 36 h.

## 3. DCN Material Systems Capable of Welding

In the past two decades, many research works have been explored around the design and fabrication of DCNs. Distinct types of traditional thermoset polymers can be converted into dynamic cross-linked networks by introducing different functional groups [98]. These DCNs demonstrate extra properties beyond traditional thermoset networks, such as weldable, reprogrammable, healable, reprocessable, and recyclable, which can be fabricated by different polymerization strategies and functional ingredients. Therefore, one potential method to realize the welding of cross-linked polymers is embedding dynamic characteristics into networks. Many resins have been successfully modified into dynamic cross-linked networks, such as epoxy resins [99], polyacrylate [28,29], polycarbonate (PC) [100], polydimethylsiloxane (PDMS) [101], poly(methyl methacrylate) (PMMA) [102], polybutadiene (PB) [26,27], polyurethane (PU) [12,23,24,25], polythiourethane (PTU) [103], etc.

Indeed, different material systems have respective suitable polymerization strategies. Herein, we focus on the most common material systems based on different initiation ways of network polymerization.

### 3.1. Epoxy DCNs: Typical Class of Thermo-Polymerization

Epoxy resins are readily synthesized by thermo-polymerization, which has been widely applied to industrial applications in modern society. Moreover, epoxy DCNs are the most studied dynamic systems due to available reagents, mature production technology, and practical industrial value. For epoxy DCNs cured by acids/anhydrides, the classical preparation strategy is thermo-polymerization and then processing by hot-pressing [104]. Therefore, it is necessary to summarize the classification and preparation of epoxy DCNs based on distinct network rearrangement mechanisms.

Epoxy DCNs based on transesterification reactions are the most in-depth studied vitrimers. Because the acids/anhydrides, as common types of curing agents, and intrinsic epoxide groups undergo transesterification at specific conditions. Usually, factors that fabricate this type of epoxy DCNs include small molecules with epoxy groups or epoxy prepolymers, acid/anhydrides, catalyst (for the majority system, is necessary to accelerate reaction rate), and specific thermal stimulus. As mentioned above, the first reported soft epoxy vitrimer was based on the reaction of diglycidyl ether and dicarboxylic acid via adding Zn(acac)_2_, which produced networks with hydroxyl groups and ester bonds [5]. Subsequently, hydroxyl groups and ester bonds realized the transesterification promoted by a proper catalyst and elevated temperature. In other words, topology rearrangement happened.

The commonly used material with epoxy groups is diglycidyl ether of bisphenol A (DGEBA). The curing agent is preferred to have two or more carboxyl groups of fatty acids/anhydrides, providing sufficient transesterification reaction sites [104]. Generally, the selected catalyst is suitable for both curing and RERs, which mainly include metal-containing compounds (especially zinc(II) salts [78,80]), and organocatalysts (especially triazobicyclodecene (TBD [79,105]). However, the presence of a catalyst also brings out some problems, which will be discussed further in Section 4.1. Therefore, some catalyst-free epoxy DCNs based on transesterification were designed [81,106]. However, constant high temperatures are essential for topology rearrangement based on transesterification, which has effects on epoxy DCNs. Generally, the external stimulus becomes moderate via doping functional particles with a photothermal effect [79,107] or magnetic response [38,108].

In contrast, epoxy DCNs containing dynamic disulfide bonds can rearrange topology under milder conditions with proper catalysts, even in the absence of a catalyst. For the preparation of these networks, dynamic disulfide bonds are always provided by disulfide-containing cure agents, such as thiols [109,110], acid [38,111], and aromatic diamine [112,113]. For example, Liu et al. [112] synthesized a biobased epoxy DCNs by curing epoxidized soybean oil with 4,4′-diaminodiphenyl methane. The catalyst-free welding process of the network based on the aliphatic disulfide linkage exchange reaction was achieved at 120 °C. Moreover, dynamic disulfide bonds in some systems could respond to light stimuli with specific light wavelengths [113], which can realize photo welding.

Dynamic imine bonds are also frequently introduced into epoxy networks which undergo distinct rearrangement mechanisms and multiple stimulus-response. As Table 1 shows, C=N bonds of an imine of Schiff base experience reversible breakage and reformation through imine condensation, amine–imine exchange, and imine metathesis, which are associated with dissociative and associative mechanisms. Usually, imines are fabricated by a condensation reaction between an aldehyde/ketone and a primary amine [104]. Moreover, for the preparation of epoxy DCNs containing imine bonds, the reactants with primary amine groups, epoxy groups, and formyl groups are always indispensable [104]. For example, by reacting with amino-terminated Jeffamine and epoxy prepolymer containing Schiff base bonds provided by Vanillin and 4-aminophenol, Zhao et al. [88] synthesized epoxy DCNs bearing aromatic imine bonds. The multifunctional DCNs demonstrated degradation and recycling in the presence of solvents via a dissociative mechanism, and water-driven malleability and thermal-induced weldability via an associative mechanism.

Epoxy DCNs based on dissociative reaction are widely prepared as well. In particular, epoxy DCNs based on DA reaction are the representative dissociative class. In 2006, Liu et al. [114] synthesized epoxy DCNs based on DA reaction in two steps. At first, maleimide and furan monomers were fabricated using epoxy compounds, and then thermally mendable epoxy-like DCNs were obtained. However, this synthetic route of incorporating DA groups into epoxy monomers is too expensive for widespread promotion [69]. Therefore, many efforts have been made to introduce DA groups into cross-linkers rather than epoxy monomers [69,70,71], whose ingredients always include commercial epoxy monomers, furans containing primary amine (commonly furfural amine), and bismaleimides. Nevertheless, some problems still need to be improved in the preparation strategy, such as simplifying the complicated synthesis process and preventing undesirable side reactions.

In contrast to epoxy DCNs based on reversible covalent bonds, dynamic epoxy systems with non-covalent interactions have been less reported in the literature containing hydrogen bonds [57,115,116], host–guest chemistry [53,54], etc. In 2015, Federica Sordo et al. [57] designed dynamic cross-linking rubbers based on hydrogen bonds using DGEBA and TGMDA epoxy resins, which possessed excellent self-healing properties but demonstrated relatively poor mechanical strength. This system and most epoxy networks with non-covalent interactions showed weak mechanical performance and/or low T_g_s [63]. Thus, they hardly withstood large loading, resulting in some limitations on structural applications [63,104]. In recent years, researchers have focused on developing epoxy DCNs with comparable performance from renewable resources, such as epoxidized soybean oil (ESO) [112,117], glycyrrhizic acid (GL) [117], vanillin [88], and epoxy natural rubber [35]. In 2020, Wu et al. [117] fabricated fully biobased and weldable DCNs from ESO and natural GL, which avoided the use of nonrenewable petroleum resources. This combination of green engineering and green chemistry concepts could reduce the negative impact on the environment caused by the chemical industry.

Generally, thermo-polymerization and hot-pressing are the most common preparation strategies compared with photo-polymerization [53,118] and solution polymerization [111,119] for epoxy DCNs. This polymerization route demonstrates many merits, such as ease of operation and solvent resistance [104], whereas processing through hot pressing is challenging to manufacture elaborate and precise structures. In other words, chemical welding caused by hot pressing is easy and feasible for simple rectangular structures with appropriate contact areas.

### 3.2. Acrylate-Based DCNs: Typical Class of Photo-Polymerization

Acrylates are common among the most reactive monomers polymerizing via photoinitiation and chain-growth reaction. This characteristic allows UV-curable acrylate-based resins to be compatible with light-based 3D printing technologies, such as stereolithography (SLA) and digital light processing (DLP). Complicated and individually shaped architectures with a high resolution can be obtained in the printing process, in which polymerization and forming are carried out simultaneously. Hence, recent attention has been directed toward developing a novel synthesis strategy incorporating dynamic bonds into photo-3D printing resin formulation [120]. Consequently, many efforts have been made to adjust and improve printing precursor ingredients and the mutual effect. Typically, photo-curable dynamic acrylate resin formulation components include acrylate monomers or oligomers, cross-linkers, photoinitiators, and photoabsorbers. In addition, catalysts accelerating RERs are also indispensable for some associative printing precursor systems.

Due to intrinsic hydroxyl esters, acrylate DCNs based on transesterification are the most studied acrylate DCNs. In 2018, Zhang et al. [28] realized high-resolution DLP-printing of acrylate DCNs with transesterification, in which specific 3D-printed objects could be welded together at 180 °C via the catalysis of Zn(acac)_2_·*x*H_2_O. This pioneering work has contributed significantly to the development of photo-curable acrylate DCNs. However, organic zinc salts are poorly soluble in the majority of typical acrylate monomers. Subsequently, Gao et al. [58] reported another photo-curable acrylate-based formulation containing TBD as a transesterification catalyst, while the manufacturing process had a relatively slow speed. The reason was that TBD could retard radically induced photopolymerization reactions as radical scavengers [121]. While common transesterification catalysts could be added to UV-curable acrylate formulation, some questions remain, such as the weak solubility, slow cure rate, and the short pot life of acrylate resins [121]. So, further research has been studied in which the focus is placed on balancing the catalyst and photo-curable system to overcome the mentioned problems.

In 2021, Sandra Schlögl and his group reported a series of systematic formulation studies around thiol-acrylate-based photopolymers and acidic organic phosphate catalysts, showing excellent compatibility and storage stability [61,121,122,123,124]. Some organic phosphates have been selected, such as methacrylate phosphate (Miramer A99) and triphenylsulfonium phosphate, which are ideal transesterification catalysts for developing photocurable acrylate DCNs as they do not compromise on cure kinetics of photocurable acrylate and thiol-acrylate systems. Moreover, the introduction of thiol-cross-linkers could follow a mixed-mode photo-polymerization mechanism involving both chain-growth (homo-polymerization) and step-growth polymerization reactions (thiol-acrylate addition reaction) [121] (Figure 2). This strategy has combined the advantages of dynamic covalent networks with the salient features of click chemistry, such as homogenous network property and low shrinkage stress [61].

Extensively, catalyst-free acrylate-based DCNs with other dynamic mechanisms have been DLP-printed as well, such as disulfide exchange [125], dynamic imine exchange [126], dynamic urea exchange [46,127], boronate ester exchange [85], and DA reaction [128,129]. In addition, some research has also been carried out around hybrid mechanisms including non-covalent interactions [22,125]. For example, Li et al. [125] DLP-printed dynamic acrylate networks cross-linked by disulfide bonding and hydrogen bonding endowed printed objects with excellent thermal weldability. In addition, acrylate-based DCNs fabricated from biobased ingredients also attracted increasing attention in recent years [126,130,131]. Normally, the acrylate-based DCN studies mentioned above have shown successful photo-curing and subsequent reprocessing, which typically demonstrated repairability but could be further adapted for the chemical welding of networks.

## 4. Chemical Welding of DCNs

In recent years, a constant focus has been placed on welding among cross-linked network polymers. One novel approach to realize this process is introducing dynamic bonds into cross-linked networks, which achieves assembly or repair via topology rearrangement under external stimuli. For welding progress, reversible chemical interaction and physical form combination need to be coordinated on both the time and space scale. In other words, the parts to be welded need to contact each other firstly, which could be assisted by clamps or weights. Subsequent reversible interactions at the contact interfaces are completed within an adequate time, during which the welding process is induced via external stimuli. Usually, familiar external stimuli include temperature, light, pH, solvent, electricity, and magnetic field. Herein, we place the main focus on chemical welding via DCBs.

### 4.1. Thermal-Induced Welding

Thermal stimulus is the most common trigger to realize the chemical welding of DCNs. Besides individual DCNs, the thermal-induced technique is also applied to DCN composites. The manipulation is easy; it only overlaps or stacks two DCN objects and then heats the joining region [76]. Generally, the welding performance is evaluated by mechanical property testing, that is, the typical lap shear test and tensile test. For some overlap welded film samples (thickness is less than 1 mm), the lap shear test is also known as the uniaxial tensile test in some studies because the virgin and the welded samples possess similar dimensions. In addition, some research also demonstrates the joining process using head-to-head welding or by connecting the cross-sections of the cutting samples, which compares the mechanical properties before and after welding by tensile tests. Besides intrinsic system properties, the joint strength strongly depends on processing parameters, including welding temperature, welding time, and welding pressure.

The influence of processing parameters on welding interface strength could be predicted in advance by the molecular dynamics simulation (MD simulation) technique. Qi et al. [132] presented a multiscale modeling framework to study the thermal-induced welding behavior of epoxy DCNs based on transesterification, which described chain density evolution via changing external factors to predict assembly interfacial mechanical properties. Specifically, welding pressure only affected the entire contact area between two parts, and further interfacial chain density increment on the total contact area was dependent on the welding temperature (T) and welding time (t).

The mentioned parameters (T, t) are closely related to the RER kinetics within the thermal stimuli, in which the kinetics are strongly controlled by the type [80,122,133] and concentration [21] of the catalyst. Therefore, the catalyst is a likewise important factor affecting the welding process. Mathieu Capelot et al. [21] experimented with distinct welding performance through epoxy/acid DCNs (T_g_ ~ 15 °C) by adjusting the transesterification catalyst concentration. The lap shear test results indicated that the sample assembly containing 1 mol% Zn(OAc)_2_ broke with a force of ~15 N, and the welding proceeded after one hour at 150 °C. On the other hand, an assembly catalyzed at 5 mol% broke at 27 N under the same conditions (Figure 3). Moreover, with 5 mol% Zn(OAc)_2_, identical welding performance was observed during assembling for one hour at 125 °C or 30 min at 150 °C. (Figure 3) This work has also confirmed that the robust welding process based on RERs can be achieved in wide time-temperature windows.

For the welding process of DCNs, the selected catalyst needs to be adapted to the chemical structure of the network and the reaction type of RERs, but its existence also brings the following adverse effects [106,134]: (1) the toxicity of most catalysts will limit potential applications of welding; (2) the poor compatibility between the polymer and high loading level catalyst will affect the mechanical property of welded structure; (3) the thermal instability of welding system caused by the catalyst will accelerate deformation and degradation at elevated temperatures.

Therefore, researchers have explored abundant catalyst-free welding strategies by designing structures and varying reversible interaction types. For example, Han et al. [106] synthesized weldable epoxy/anhydride DCNs in the absence of a catalyst, in which a sufficient number of free hydroxy groups and a large free volume in designed network structures sped up the transesterification. By pressing one hour at 150 °C, the assembly showed a tensile strength of ~47.3 MPa, which was almost identical in contrast to the original sample (~47 MPa). Otherwise, Yang et al. [26] realized the self-welding process based on another mechanism without a catalyst, driven by dual-dynamic units (imine and disulfide) of polybutadiene networks at 100 °C (Table 2). The welded sample could support a 1 kg weight that was 3000 times heavier than itself. Moreover, Max Röttger et al. [102] demonstrated dissimilar thermoplastic-based DCN welding via catalyst-free dioxaborolane metathesis. After a 20 min contact at 190 °C under 11 KPa pressure, the welding performance between the PMMA-vitrimer and HDPE-vitrimers was so robust that bulk fracture systematically occurred in PMMA (Figure 4) instead of within the overlapping regions.

Welding is an excellent choice for repairing damaged regions and realizing complicated structure assemblies. For example, Zhang et al. [28] demonstrated a repairing method by thermal-induced welding photocurable acrylate DCNs. As shown in Figure 5, to repair the sample with a hole, researchers first filled the hole with the DLP-printed formulation solution and then applied the UV (365 nm) irradiation to cure for 10 min and thermal treatment to trigger transesterification at the contact interface at 180 °C for 4 h. The obtained repairing sample was compared with the control sample using the tensile test to testify the repairing capability (Table 2). Fang et al. [46] realized modular 4D printing and thermal-induced assembly via interfacial catalyst-free RERs, which revealed a remarkable potential for various shape-shifting devices. Different parts were DLP printed by varying the printing formulations. Subsequently, three 4D-printed modules were welded together at 160 °C for two hours based on dynamic hindered urea linkages. The obtained assembly demonstrated sequential compression under a specific load owing to the distinct thermomechanical characteristics of individual modules. In addition, Li et al. [29] demonstrated another RER assembling strategy of complicated DLP printed modular, which needed assisted solvent(Figure 6). A drop of UV-curable recycling (UVR) solution containing potential dynamic characteristics was added and UV-cured between neighboring grids to be joined, and then a thermal-triggered interfacial transesterification resulted in the welding assembly [29]. Different from the above works, Zhu et al. [22] assembled differential grids using non-covalent bonds. DLP-printed acrylate-based DCNs could be tuned from soft elastomers to rigid plastics, whose assembly was obtained via dynamic ionic bonding and hydrogen bonding at 90 °C for 12 h. (Figure 6). These contributions allow an extension to fabricate multi-material 3D devices with multifunctions beyond shape-shifting. Desirable structures could be built by bonding different units, which breaks the limitation of printer size and targets complicated configurations.

In addition, the preparation and welding of DCNs through supramolecular interactions have also been researched [22,57,146,147]. Most of these DCNs are based on soft networks, which show low T_g_ (<50 °C) and strength (<10 MPa) compared with CANs [2]. However, it is also a possible way to realize welding at ambient temperatures without any stimuli [147].

Compared with the single chemical bonding, the combination with physical entanglement can achieve different extents of welding between systems, depending on the relationship between the welding temperature and the characteristic temperatures of the linear chain and the cross-linked network in the blending DCN systems. Chen et al. [147] presented a work in which the welding mechanism and condition were gradually varied with changing compositions in blending epoxy DCN systems. These systems were fabricated by mixing prepared epoxy vitrimers and thermoplastic TPU (melting temperature T_m_ = 155 °C) at different ratios, which successfully self-welded. The welding mechanisms ranged from mild hydrogen bonding to thermal-induced hybrid interaction. For the welding process based on the latter mechanism, contacted systems have achieved both transesterification and entanglement of TCL chains, at which the welding temperature exceeded the melting point of TCL. This work showed that the welding of cross-linked networks could also be achieved via the co-action of chemical bonding and physical entanglement. However, it was challenging to distinguish the contribution of chemical or physical processes because both acts were developed with elevated temperatures. Subsequently, Ji et al. [141] prepared multi-shape memory epoxy polymers through a thermal-welding strategy based on rapid disulfide linkage exchange and entanglement of PCL chains across the contact interface. The multilayer system could be programmed at elevated temperatures and underwent gradual overall recovery with the dropped temperatures, fabricated by stacking layers with respective glass transitions together at 130 °C for 60 min (Figure 7). The key factors of the stacked multilayer system to realize the multi-shape memory effect are the separation of T_g_s and strong adhesion among layers.

In contrast to stack welding, the overlap welding strategy enables the shape of each part to be programmed and recovered independently at corresponding transition temperatures [141,148]. Liquid-crystal elastomers with exchangeable links (xLCEs) show great potential in shape-memory actuators under different stimuli because of their intrinsic several characteristic temperatures [149], such as T_g_, T_v_, and isotropic transition temperature T_i_. In 2016, Pei et al. [148] realized the regional shape control of multi-shape memory xLCEs using an overlap welding strategy. Three epoxy vitrimers (BA, MS, and BP) were synthesized with different thermomechanical properties, and both MS and BP belonged to smectic liquid crystalline networks. Three strips were overlap-welded based on RERs by three vitrimers in different orders at 130 °C, and then all strips were reprogrammed into the same 3D spiral shape (S1). However, subsequent recovery occurred in totally different sequences and finally recovered the same initial shape (Figure 8). In summary, although it was the same spiral-to-straight change in the same thermal environment, different recovering processes could be realized for the demand based on the designed region arrangement through overlap welding, and each section could respond independently under the corresponding stimulus. Moreover, this strategy made it possible to combine multiple responsive materials. As shown in Figure 8, the multifunctional strip could respond to heat and light, and different functional fillers were doped into the BA network.

Thus, networks that introduced additional fillers bring out extra functions and enhanced properties beyond conventional individual DCNs. Some of them still could be assembled by thermal-induced welding strategies, such as reinforced fibers [135,142,143], carbon nanodot (CD) [150], carbon nanotubes (CNTs) [69,113,115], silicon [82], iron oxide [36,37,38], silver nanowires [22,110], gold nanoparticles [151], oligoanilines [15,34,35,144], and CH_3_NH_3_PbI_3_ [152]. However, the introduced particles may aggregate, resulting in inhomogeneous and fragile networks and unsatisfactory topology rearrangement. So, it is vital to balance filler-enhanced properties and network plastic-processing ability. Tang et al. [150] introduced carboxyl group-functionalized carbon nanodot (CD) into epoxidized natural rubber (ENR) to prepare weldable DCN composites, where transesterification could occur in the ENR-CD interphase (Figure 9). Additional CD played an essential role as not only a cross-linker and enhancer but also a transesterification catalyst that worked together with Zn(Ac)_2_. The samples containing 4.68 wt% CD were welded together by hot pressing at 180 °C for 10 min.

Welding of networks with high-addition content has also been reported. Legrand et al. [82] achieved welding of epoxy DCN composites within 40 wt% of epoxide-functionalized silica. In contrast to common hydroxyl-functionalized silica, epoxide-functionalized silica could exchange β-hydroxyl ester bonds with chains to improve their dispersion state in the network. Figure 9 performs on lap shear test results of assembly welded at 190 °C by applying a ~30% strain for various welding times. After 30 min, the rupture occurred in bulk rather than welded interface, with a fracture stress of a lower value (~52 N). Chabert et al. [135] realized the joining of epoxy composite materials containing 53 vol% of reinforcing fibers, and multiple welding was performed at 160 °C for 90 min under 1.7 MPa or 4.4 MPa. However, it always broke at the overlapped region during stretching because of the discontinuous fibers at the interface. Overall, improving poor compatibility could become a breakthrough in weldable composite systems with a high weld performance.

A weldable network system containing conductive fillers can be used in flexible device fabrication. In 2018, Deng et al. [110] reported a self-weldable triboelectric nanogenerator (TENG) by doping a silver nanowire percolation network into a vitrimer elastomer based on an aliphatic disulfide bond. Figure 10 shows the separated TENG rejoining process following heat treatment at 65 °C for one hour. The attached silver nanowires were also welded together simultaneously, further recovering the original connectivity. In addition, a terrace-structured VTENG was designed with higher conductive efficiency by flipping adjacent blocks, clipping them together, and then heat-welding. This work presented self-weldable TENG with excellent design flexibility that can be applied to electronic sensors.

Most of the welding property research has been carried out on a macro scale. In contrast, there has been little attention to the micro variation in the welding seam. However, the reported simulation research [153,154] and practical measurements [155,156] of microcosmic interfacial broadening caused by reversible reactions through activated atoms have almost been based on a thermal stimulus. In 2016, Yang et al. [153] used MD simulation to investigate the welding process of active atoms crossing interface by tracking the trajectory of activated atoms across two same epoxy DCN systems. The predicted simulation results revealed that the welding system could fully reach the same modulus and yield stress as a fresh network under a sufficient processing time. In 2018, He et al. [155] visualized the interfacial broadening width between two epoxy DCNs with distinct moduli via atomic force microscopy nanomechanical mapping (AFM-NM). As the exchange reaction went on, the welding zone showed a gradient in the concentration, which could quantify the kinetics of the exchange reaction. Moreover, this technique simultaneously provided information including the variable modulus and composition across the interface caused by segment exchange (Figure 11), extending a new pathway to calculate the topology arrangement activation energy of vitrimers.

### 4.2. Photo-Induced Welding

Photo-induced welding is a method of performing material joining through the dynamic interaction of cross-linked network systems under light stimulation. In contrast to direct heating, light is more flexible, effective, moderate, and easy to control. Irradiation can be controlled in selected areas to avoid unnecessary impact on surrounding areas [136]. Familiar light stimuli include ultraviolet (UV) light [157,158], visible light [24,60], near-infrared (NIR) light [53,107,113,137], etc. Among them, the most used laser wavelength is 808 nm. The photo-induced technique is successfully applied for bonding DCN systems within intrinsic photo-responsive dynamic bonds and/or additional photothermal effect components, such as trithiocarbonate-based individual DCNs and DCN composites containing CNTs.

Disulfide compounds can undergo bond exchange in the presence of a catalyst [159] or under UV stimuli [160] but usually occur within the environments of solvents and inert gases [24]. In 2011, Yoshifumi Amamoto et al. [158] reported photo-induced welding of acrylate DCNs at room temperature, in which the homolysis of C-S bonds triggered by UV irradiation resulted in a reshuffling of trithiocarbonate (TTC) under the catalysis of radicals. The pieces to be welded were placed under mild pressure by a 4 g weight in an acetonitrile solvent (to improve the movement of the chain) and under nitrogen (to protect C-based radicals). After UV irradiation (330 nm) for 8 h, the welded sample showed a tensile modulus value of (65 ± 11) KPa, whereas that of the pristine sample was (69 ± 6) KPa. Moreover, the authors of [24] subsequently reported another photo-induced welding process based on the reshuffling of thiuram disulfide (TDS) units at more moderate conditions [24]. The samples were exposed to visible light (a tabletop lamp) for 24 h to achieve assembly in air at room temperature, without a solvent. This rearrangement process was achieved by degenerative radical transfer and radical crossover reaction. Testing Young′s modulus value of the welded sample was (612 ± 152) KPa, whereas that of the original sample was (544 ± 208) KPa. However, such welding processes had to be conducted on freshly cut samples. Otherwise, the radicals required for the welding may decompose or diffuse into the sample.

Compared to the photo-weldable networks based on radical-exchangeable reactions, a new strategy without sensitive radical participation showed superior photo-weldability [60]. Indeed, the process of sunlight-driven welding of PU DCNs was achieved via dynamic aliphatic disulfide bonds, hydrogen bonding, and entanglement of PEG chains [60]. Two dumbbell-shaped film specimens were brought together to integrate under sunlight for six hours. Interestingly, in this welding process, the unique DCN chemical structure is also of importance. The network (T_g_ = 29.8 °C) was a glassy state at room temperature, which turned into an elastomeric state with the elevated temperature by sunlight and enhanced the movement of lineal PEG chains at the contact interface. Accelerated physical interdiffusion and entanglement between two pieces would again promote dual chemical bonding acts.

Faster welding studies have been reported, with several minutes, even seconds, by fabricating dynamic network composites with a photothermal effect. For example, Guan et al. [113] introduced CNTs into the dynamic network to realize photo welding. Two systems were prepared: epoxy-based DCNs containing dynamic S-S bond (SH) and SH/CNT, which introduced 2.5 wt% CNTs into the SH network. Samples of two systems were welded under NIR irradiation for 60 s, and distinct phenomena were observed. The SH system reached 130 °C in 60 s upon NIR irradiation, and the tensile strength of the welded sample could recover ~85% of that of the original SH sample (1.7 MPa). While the SH/CNT system increased rapidly to 160 °C at the same condition, assembly tensile strength recovered ~90% of the original SH/CNT sample (2.6 MPa). These results suggested that the introduction of CNTs enhanced the photothermal conversion efficiency of the system, resulting in higher temperatures leading to faster network rearrangement furthering higher welding performance for the same joining time.

Infrared light has a strong penetration ability, which is conducive to developing unique welding technology, such as transmission welding [107]. Yang et al. [107] realized transmission welding of different materials under IR irradiation (808 nm, 0.84 W/cm^2^). Epoxy DCN composites based on transesterification (CNT-vitrimer) were prepared by doping 1 wt% CNTs into epoxy vitrimer (Table 2). The transmission welding was finished after 60 s between CNT-vitrimer and ordinary thermosetting epoxy cured by diacid or anhydride, and the transesterification catalyst in the CNT–vitrimer side promoted topology rearrangement at the welding interface with the thermal transmission of CNTs. Moreover, thermoplastic PE was easily joined with CNT-vitrimer by IR light because the local welding region temperature was around the melting point. Two lap shear test results indicated that the tensile strength values of welded samples were between that of two individual materials (Figure 12). After that, the researchers also applied this local and effective strategy to epoxy LCEs [136]. The rapid photo-welding process could be completed in seconds to avoid the disappearance of desirable alignment, which was difficult to solve under the constant thermal stimulus.

In addition to CNTs, other light-responsive components have also been introduced into polymer networks to bring out multi-stimuli responsivity and multi-functionality [34,35,137]. Li et al. [34] doped organic polydopamine nanoparticles (PDA NPs) into epoxy LCEs to realize heat-welding and photo-induced shape programming within seconds via the photothermal effect of PDA NPs, which fabricated dynamic 3D structures using 2D heterogeneity films (Figure 13). In 2017, Chen et al. [137] used a synthesized amino-capped aniline trimer (ACAT, an oligoaniline) as an alternative addition to responding to photo stimuli. The prepared ACAT-vitrimer films were joined together under IR irradiation (808 nm, 0.7 W/cm^2^) for 30 s. Moreover, this epoxy DCN system, in the presence of ACAT, could not only exhibit heat- and light-responsiveness but also respond to pH, voltage, metal ions, and redox chemicals, making a smart polymer with multi-functional properties. In 2018, Hu et al. [53] modified graphene oxide (GO) and copolymerized the obtained β-cyclodextrin/graphene and unsaturated epoxy resin. The epoxy supramolecular network could realize the self-welding of damaged regions under thermal stimulus (120 °C) or NIR irradiation (2 W) via host–guest interactions.

### 4.3. Solvent-Assisted Welding

Solvent-assisted welding of DCNs proceeds within a solvent medium, where the participating solvent improves the migration ability of swollen chain segments and accelerates their interaction at the contact interfaces. This strategy broadens the application scenarios for welding because it is capable of performing in environments where it is difficult to provide thermal stimulation. However, it usually needs elevated temperatures to shorten the welding time. Two ways to introduce a solvent exist in the literature: (1) dropping into the local welding area precedes overlapping dry parts; (2) immersing the whole part precedes touching wet samples. In the latter method, soaking time and temperature are non-negligible process parameters, and the swollen degree of the sample, even the depolymerization degree, will affect the subsequent welding process. Commonly, solvent welding achieves joining between individual networks, and typical mechanisms include dynamic imine reaction and transesterification.

Given that topology rearrangement of the polyimine synthesized by diamine and dialdehyde could be activated by water [33], Albert Chao et al. [161] investigated the effect of solvent polarity on swollen imine cross-linkage networks fabricated by diamine and trialdehyde with an equal molar ratio of amine and aldehyde functionalities. The rheological measurements revealed faster dynamic imine RERs in the polar solvents (the relative imine exchange rate in polar solvents: acetonitrile≫DMF>toluene) than in non-polar solvents. Coating a little acetonitrile solvent on the cut surface allows two fracture imine DCN objects to be welded at room temperature. Otherwise, the RER kinetics test suggested that RER was significantly accelerated by the presence of primary amine compared with water, which showed that the imine bond exchange is mainly catalyzed by the primary amino functionalities.

Lei et al. [31] also observed similar phenomena by welding poly(imide-imine) hybrid DCNs (PIIH) using different solvents. Mechanically robust PIIHs (T_g_ = 217 °C) were synthesized by hybridizing rigid imide moieties and dynamic imine linkages. One drop of DETA solution in DMF/acetonitrile solvents (*v*/*v* = 1:1, 10 mg/mL) was added to the welded region, followed by hot-pressing the entire assembly (Table 2). The manufactured assembly always broke at the bulk material instead of within the overlapping region. However, the welding performance was unsatisfactory under the same conditions when DMF or water acted as the assisted solvent. In general, unlike the structure of pure polyimine, the RERs of this imide-imine hybrid DCNs required the catalysis of amine (Figure 14).

The solvent-assisted strategy is also employed in the joining of epoxy-based DCNs. Liu et al. [86] added one drop of diethylenetriamine/DMF solution (10 mg/mL) to the welded position, pressed two epoxy-DCN parts with a clip, and heated under 140 °C for one hour. The welding process depended on the imine exchange reaction provided by the synthetic dynamic imine cross-linker. Moreover, the welded sample can sustain a tensile stress of 63.89 MPa (Figure 15), which is 87.8% of that of the virgin sample. Otherwise, the SEM images of the overlapped region after welding showed an excellent connection between the two objects.

All the mentioned assisted solvents participated in the welding processes in the first method because imine bonds in the presence of just a small amount of amine could undergo degenerative bond exchange rapidly. In contrast, transesterification requires a higher level of catalyst content, but the movement of swollen chains and exchange speed between bonds are relatively unsatisfactory. Hence, the solvent welding processes based on transesterification usually adopt the second method: all the workpiece is soaked in the solvent in advance. For example, Yang et al. [83] immersed epoxy-DCN films containing TBD into pure THF solvent till they became fully swelled, and then the THF evaporated, maintaining good contact between the two films. The welding was finished after the solvent achieved complete evaporation. Moreover, Figure 16 depicts the heterogeneous film assembly under the same conditions. The entire welding process was carried out at room temperature, whereas the complete evaporation of THF took three days. Further, Zhang et al. [139] realized the welding of normal thermoset epoxy networks in this way, in which TBD catalyst, as a solute, was added to the presoaking solvent THF. The catalyst molecules infiltrated into the network with the solvent to endow thermoset weldability. Meanwhile, the resulting network mechanical property has been enhanced due to the elaborately designed structure (Table 2). Moreover, an extended phantom network model was established to deeply investigate topology on interfacial welding and self-toughening behaviors.

Ethylene glycol (EG) is often used as a reactive solvent in the welding process based on transesterification. The hydroxyl groups in EG could react with the intrinsic ester groups in the networks at a specific condition, and EG has an appropriate boiling point (197.3 °C). In 2016, Shi et al. [84] chose EG as a presoaking solution to achieve pressure-free epoxy DCN welding. First, they immersed the epoxy DCNs into EG solvent at 180 °C for 30 min; then, the samples were taken out of the solvent and stacked together in a room environment; finally, welding was achieved at 180 °C for 180 min in an open-air environment. It can be seen from Figure 16 that almost perfect welding was carried out in which the roughness of the presoaking sample surface decreased and resulted in good contact. Furthermore, the capillary force also contributed to the mentioned process caused by the partial dissolution in the presoaking step. Hence, it is essential to systematically study the presoaking condition because oversoaking results in assembly deformation.

Subsequently, Shi et al. [138] adopted a similar strategy to weld normal anhydride-cured thermoset epoxy and optimized proper process parameters, including soaking temperature and soaking time, in which the EG presoaking solution contained transesterification TBD. A gentler welding process was finally determined: immersed thermoset parts into EG solution containing 1 wt% TBD at 80 °C for 120 min; after wiping off the solvent on the surface, welded at 100 °C for 80 min under 0.5 MPa. In addition, the theoretical model was established to predict the lap shear strength under different conditions, which guided the selection of the optimum technological conditions for practical engineering welding applications.

In addition, some types of dynamic interaction could respond to pH, such as the thiol-Michael reaction [162], imine condensation [163], and metal–ligand coordination [52,164]. Such pH-responsive dynamic bonds enable welding for particular scenarios, such as undersea water welding. Xia et al. [52] designed hyperbranched polyurethane DCNs containing catechol-Fe^3+^ coordination bonds capable of repeated self-welding in seawater without manual intervention (Figure 17). Two pieces were clamped together in artificial seawater (pH = 8.3) at 25 °C for 24 h (Figure 17). The welded sample showed a tensile stress of ~2 MPa, which is 87.2% of that of the virgin sample. Subsequently, the authors also synthesized another polyurethane DCN containing the reversible coordination bonds of catechol-B^3+^ to achieve underwater welding in the broader pH range, including alkaline and neutral conditions [164]. These works have contributed to expanding the adaptability and application range of underwater rejoining polymers.

### 4.4. Electrical/Magnetic-Induced Welding

Familiar stimuli also include the role of electricity and magnetism. However, individual network polymers hardly respond to these two types of stimuli. The network system capable of welding under electrical/magnetic stimuli must introduce additional components with a Joule-heating effect [77,165] or magnetic response characteristics. Note that different magnetic responsive behaviors strongly depend on different kinds of doped magnetic particles and external magnetic fields. For the alternating magnetic field (AMF), magnetic particles always respond to the magnetothermal effect, while external static magnetic fields directly guide the dipole attraction between parts. At present, common additions in polymer networks with Joule-heating effect include CNTs, but magnetic responsive particles involve variable kinds of iron oxides, especially Fe_3_O_4_.

He et al. [165] exhibited electrical-induced interfacial welding of polyimine composites based on dynamic imine exchange caused by the Joule-heating effect. The composite system (T_g_ ~ 50 °C) was direct ink writing printed with dynamic C=N bonds on the chain backbone and conductive MWCNT additions. Two cut samples were brought together, and a direct current with 0.15 A was applied to achieve interfacial welding (Figure 18). Due to the Joule-heat effect, the temperature of the welding interface rapidly saturated around 63 °C in several minutes, enough to trigger the RERs. After 40 min, the assembly had almost the same conductivity property as the fresh sample (2.7 s/cm). After 50 min of welding, the strength saturated at ~57 Mpa, which was ~70 MPa for the original sample. After cutting, reduced strength may be caused by interfacial voids, which need higher pressure to improve the total contact area. In 2022, Sang et al. [77] also demonstrated a similar welding strategy based on dynamic Diels–Alder networks containing branched MWCNTs (b-CNTs). The dumbbell samples were healed through direct current voltages (24 V or 16 V) for 10 min reaching the homogeneous temperature of 100 °C at the entire area by the Joule-heating effect, triggering the DA bond re-formation (Figure 18). Note that, for electrical-induced interfacial welding, this strategy cannot realize non-contact welding because a closed-loop circuit is of the essence.

By contrast, welding under magnetic stimulus can realize non-contact processing, even remotely precise control, attracting attention for actuated applications. For example, Zhang et al. [36] welded bulk PU networks via a magnetothermal-responsive solder, which could avoid severe deformation under pressure and oxidation yellowing in direct-heat welding (Figure 19). The PU CAN solder worked via transcarbamoylation under the magnetothermal conversion of Fe_3_O_4_ nanoparticles (~20 nm) and the catalysis of DBTDL. Firstly, the welding of solder CANs under the AMF was performed for two minutes, and the lap-shear test showed that the welded solder sample possessed almost the same tensile stress as the blank sample (3.1 MPa), proving strong weldability. Next, the welding of blank PU CANs (T_g_ ~ −15 °C) via solder for 2 min in the AMF (f = 495 kHz, H = 127.331 Gs) was carried out, and the solder was heated rapidly (about 150 °C for 1 min), resulting in triggered local RERs around the contact surfaces. Subsequent lap-shear tests and swelling experiments were made to testify to the success of this welding strategy (Table 2). In 2022, Wu et al. [11] applied magnetic-assisted welding for reprogrammable soft actuators (Figure 20). This report used a series of magnetic PU LCE (M-PULCE) networks based on dynamic carbamate bonds. Moreover, multifunctional and multi-material actuators were assembled via the magnetothermal effect of Fe_3_O_4_ nanoparticles under AMFs. This welding strategy demonstrates the promising application prospect in new-generation smart devices capable of adapting functions under changing situations.

Besides the magnetothermal effect under the AMF, magnetic particles enable doped networks to mutually attract in an external static magnetic field, in which welding acts under additional stimuli. It is a promising strategy to apply to practical applications, such as magnetic soft actuators and multi-functional soft robots. For example, Mao et al. [38] reported novel strategies to realize magnetic epoxy network welding based on transesterification and dynamic disulfide exchange through multiple stimuli (Figure 21). The samples were overlapped and kept in good contact by a magnet, relying on the mutual magnetic attraction of Fe_3_O_4_ NPs (~20 nm). And then, welding was carried out within the activated topology rearrangement by heat (at 100 °C for 1 h) and light (by 808 nm NIR light for several seconds).

Innovatively, Kuang et al. [37] demonstrated remotely controllable modular assembling and reconfigurable architecture via regulable magnetization and actuation modes. In this work, every magnetic modular needed to be post-magnetized with an initial magnetization pattern before welding, consisting of NdFeB microparticles (~25 μm) and a soft dynamic network based on DA reaction (Figure 22). Firstly, a seamless welding process was demonstrated by magnetically attracted assembling modules with pre-designed configurations and subsequent heating treatment. The assembled 2D planar structures with programmed magnetization underwent complicated shape morphing upon applying a magnetic field. And then, by taking advantage of magnetic-driven remote navigation, researchers realized remotely controlled modular assembling and welding. The parts were assembled using rotating magnetic fields (3 mT, 1 Hz) and welded by a remote IR laser (1 W) for 30 s without requiring direct touch. This innovative work offers enormous potential for next-generation multifunctional assemblies and remotely controllable reconfigurable shape-morphing devices.

In addition, Dong et al. [108] reported a magnetic functional soft robot that could secure/release the object by welding/unwelding the gripper. The magnetic network-doped Fe_3_O_4_ was cross-linked by a thiol-epoxy reaction, in which welding would be performed by dynamic disulfide exchange under heat/UV stimulus. Besides gripping/releasing via welding, other functions of the robot were demonstrated under the direct guidance of a static magnetic field, such as squeezing itself to pass through an extremely confined space, reshaping, self-healing, and transporting.

## 5. Summary and Outlook

DCNs exhibited numerous unattainable advantageous characteristics for conventional cross-linked polymers, notably the ability to weld thermoset polymers. In this review, the term ‘chemical welding’ denotes the chemical bonding process occurring within the contact network under specific stimuli. This process involves the interaction of dynamic chemical bonds, leading to the interpenetration of exchanged chain segments across the interface. Various welding strategies have been elucidated in this review, but a fundamental concept remains crucial irrespective of the employed chemistry: during the welding process, local reversible interaction must occur significantly faster than overall flow or deformation. This dynamic ensures mitigation of the risk of undesirable shape loss through welding [63,166].

Welding via supramolecular interaction relies on the rapidly forming and dissociating reversible non-covalent bonds. Despite possessing highly dynamic properties without needing additional catalysts, the practical application of non-covalent welding has been constrained using relatively suboptimal mechanical properties and stability. In contrast, CANs have garnered considerable attention for developing diverse polymerization strategies and expanding multiple applications, exhibiting enhanced stability through higher bond energy [167]. The equilibrium process in dynamic covalent systems is notably slower than that in supramolecular systems, often necessitating a catalyst to regulate the reaction within the appropriate time scale. A significant amount of welding research within CANs involves using a catalyst to expedite the welding process. Nevertheless, the presence of a catalyst exerts effects on material properties and potential application. Consequently, catalyst-free welding processes have been reported and developed.

Welding strategies exhibit varying advantages and drawbacks when subjected to different stimuli. Most DCNs can be effectively assembled under thermal stimuli, encompassing both individual DCNs and DCN composites. However, prolonged exposure to heat induces alterations in the nature and configuration of the welded structure. The light-induced welding strategy, initially grounded in radical exchange reactions, is hindered by impractical conditions for widespread application. In recent years, this approach has found utility in individual networks through dynamic disulfide exchange and epoxy CAN composites within the photothermal effects. Although the photo-welding process allows precise control within minutes or even seconds, its application is limited by the low penetration of light, constraining the thickness of the welded sample. Solvent-assisted welding, employing mechanisms such as dynamic imine exchange and transesterification, facilitates the welding of conventional epoxy/anhydride cured thermosets by immersing them in solvent-containing catalysts, obviating the need for pre-fabricating dynamic networks.

Nevertheless, achieving precise local control and rapid remote operation remains challenging. Electrical-induced welding accelerates the process but falls short of achieving non-contact welding. Conversely, welding based on magnetic response demonstrates significance with its rapid response, excellent biological safety, locally precise control, and robust penetration. However, achieving uniform dispersion of magnetic particles within the network requires improvement.

While substantial advancements have been achieved in the chemical welding of DCNs in recent years, several inherent challenges and potential opportunities necessitate consideration in network fabrication, exerting stimuli, welding performance, and practical application. Firstly, prospective endeavors will consistently prioritize designing and fabricating novel bio-based DCNs characterized by environmental friendliness and sustainability. This includes the utilization of renewable and non-toxic monomers/cross-linkers, as well as the integration of catalyst-free dynamic interactions, and the incorporation of bio-compatible multifunctional additions.

Secondly, concerning the practical application of welding, a strategic focus should be placed on the amalgamation and optimization of diverse external stimuli and responsive interactions. For instance, in aerospace applications, a combination of ultrasound and microwaves demonstrated the capability to achieve precise and efficient remote control [63]. Furthermore, stress stimuli have garnered heightened attention in recent years, exhibiting the potential for network arrangement under mild conditions, such as at room temperature, through external stress [45,168,169,170]. For medicinal or biological applications, welding strategy can draw inspiration from dynamic combinatorial chemistry, employing supramolecular interactions and reversible covalent bonds to respond to mild conditions.

Thirdly, there is a need for more comprehensive microscopic and theoretical investigations based on distinct mechanisms. Presently, research on chemical welding predominantly emphasizes process parameters and macroscopic mechanical properties. In contrast, theoretical exploration of microscopic welding interface behavior remains relatively scarce [132,153,154,171,172,173]. Traditional experimental research methods face challenges in accurately and comprehensively characterizing the microstructure changes within networks and establishing a specific relationship between the microscopic morphology and macroscopic properties of welding regions. However, these challenges can be effectively addressed through theoretical simulation studies.

In summary, the weldability of DCNs has garnered significant attention due to its wide-ranging application prospects. Utilizing DCNs with epoxy CANs as an illustrative example can further facilitate the expansion of production scale and a gradual transition into industrial applications, including defense, aerospace, medicine, construction, electronics, and intelligence. Additionally, developing this assembly strategy may herald a new landscape for the composite manufacturing industry, enabling functional integration while concurrently reducing overall costs.

## Figures and Tables

**Figure 1 polymers-16-00621-f001:**
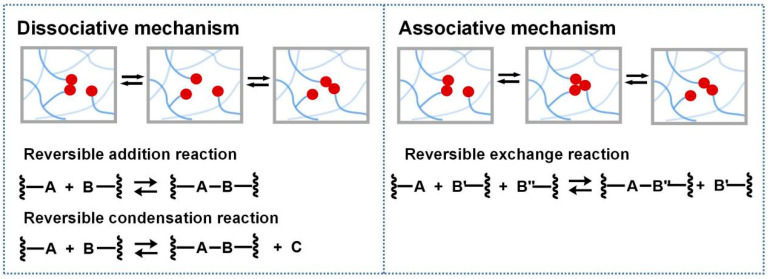
Two mechanisms for the CAN topology rearrangement.

**Figure 2 polymers-16-00621-f002:**
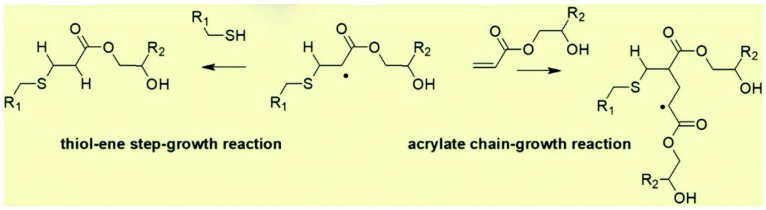
Schematic representation of the photo-curing reaction in thiol-click formulations [121]. Adapted with permission from [121]. Copyright 2024, Royal Society of Chemistry.

**Figure 3 polymers-16-00621-f003:**
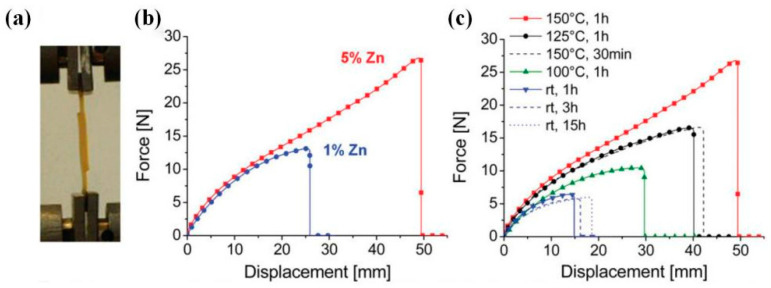
(**a**) Photograph of a sample during the lap-shear test. (**b**) Stress−strain curves of samples loaded with different catalyst concentrations welded for 1 h at 150 °C. (**c**) Stress−strain curves of samples loaded with 5 mol% Zn(OAc)_2_ welded under different conditions [21]. Adapted with permission from [21]. Copyright 2024, American Chemical Society.

**Figure 4 polymers-16-00621-f004:**
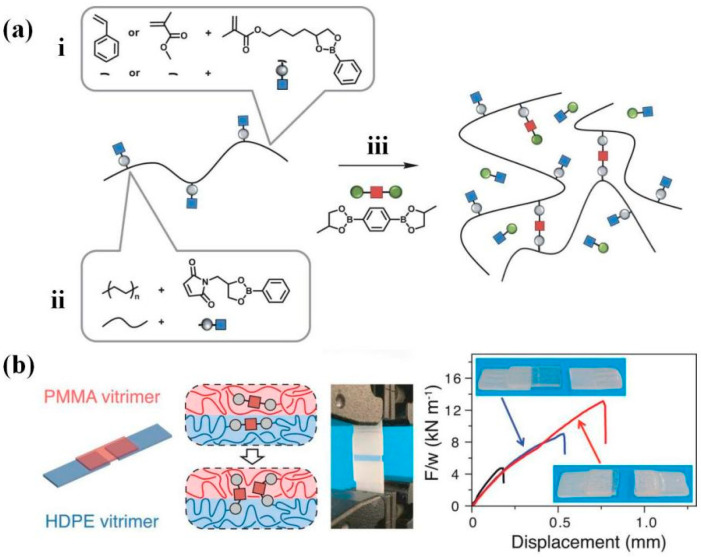
(**a**) Synthesis of DCNs. (**i**) Synthesis of copolymers containing pendant dioxaborolanes from functional monomers. (**ii**) Grafting of dioxaborolanes onto thermoplastic polymers by means of reactive processing. (**iii**) Cross-linking of functional polymers containing pendant dioxaborolane units by means of metathesis with a bis-dioxaborolane. Free dioxaborolanes formed during the cross-linking process can be kept in the system as plasticizers or removed through evaporation. (**b**) Enhanced adhesion of PMMA and HDPE vitrimers. Shown are a schematic representation, photo, and lap-shear testing of double lap joints of PMMA/HDPE dioxaborolane thermoplastic precursors (black, contact time 10 min, adhesive failure) and PMMA/HDPE vitrimers for two contact times, 10 min (blue, adhesive failure) and 20 min (red, bulk failure in PMMA) [102].

**Figure 5 polymers-16-00621-f005:**
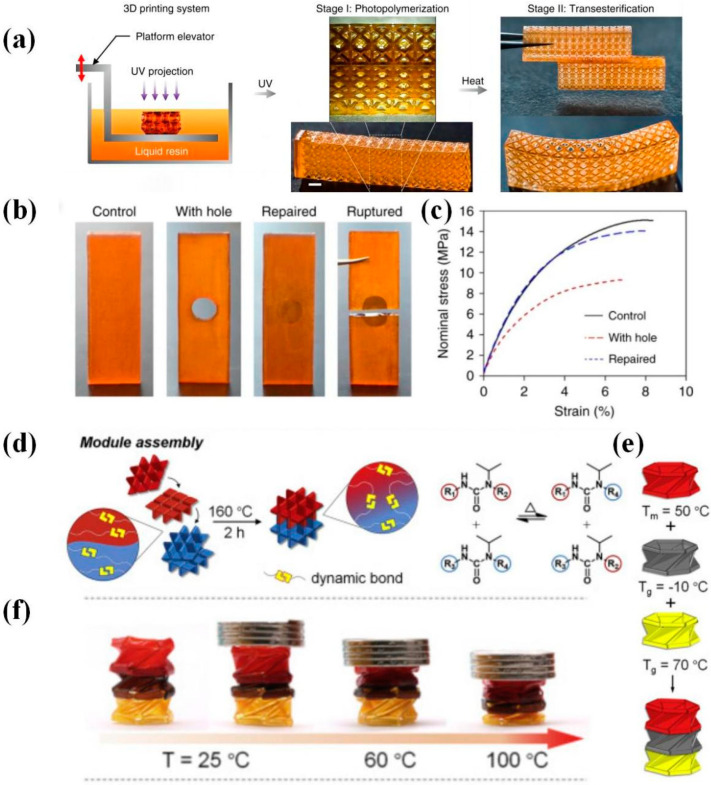
(**a**) 3D printing and assembling of acrylate thermosets. General route of 3D printing high-resolution lattice structures with a UV curing-based 3D printing system (Stage I). Two separate printed lattice structures can be thermal-welded together, and a straight lattice structure can be programmed into a bent one (Stage II). (**b**) Photos of the control sample, a sample printed with a hole, a repaired sample, and a ruptured sample after repair. (**c**) Comparison of the nominal stress (force divided by cross-section area of control sample) vs. strain for the samples in (**b**) [28]. (**d**) Module assembly: various modules are assembled by interfacial welding via dynamic covalent bond exchange. (**e**,**f**) The assembly process and deformation behavior of a Kresling-patterned multi-material cylinder [46].

**Figure 6 polymers-16-00621-f006:**
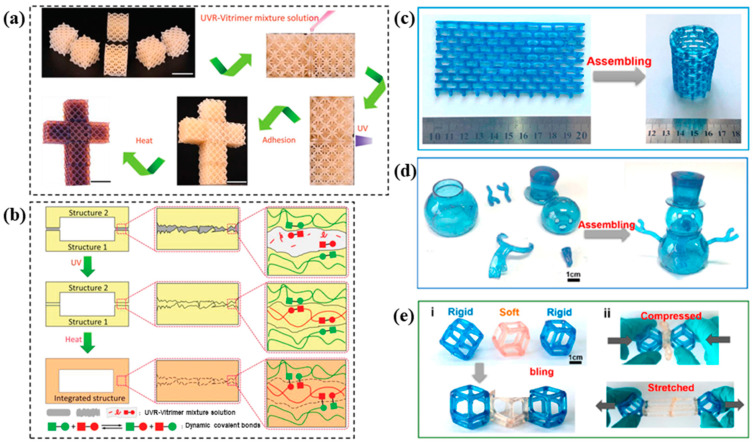
(**a**) Assembling process of multiple octet-truss structures (scale bar: 20 mm). (**b**) Illustration of the details of the assembling process [29]. Adapted with permission from [29]. Copyright 2024, Wiley. (**c**–**e**) Assembling categories based on hydrogen bonding and ionic bonding. Photographs demonstrating the assembling of (**c**) the 2D structure to the 3D structure, (**d**) small parts to a large snowman model, and (**e**) the same to different materials to prepare a complex gradient 3D structure [22]. Adapted with permission from [22]. Copyright 2024, American Chemical Society.

**Figure 7 polymers-16-00621-f007:**
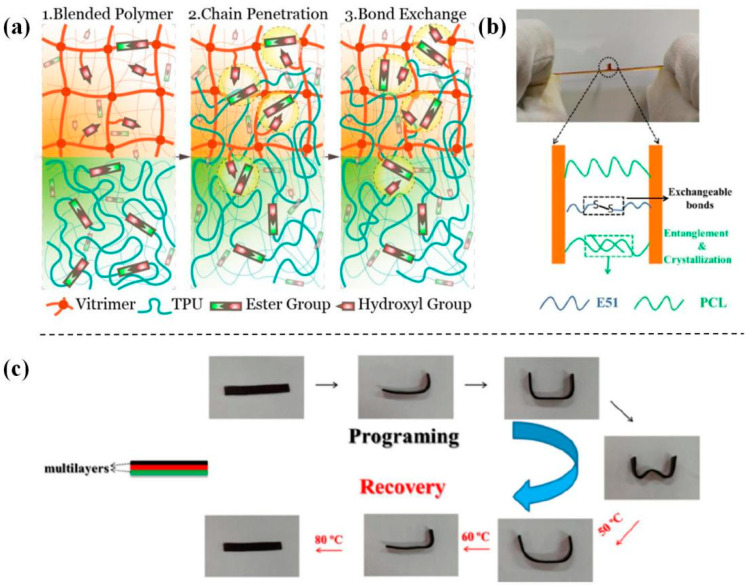
(**a**) Schematic of blending TPU/vitrimer and the mechanism of copolymerization based on the bond exchange reactions [147]. Adapted with permission from [147]. Copyright 2024, American Chemical Society. (**b**) Schematic illustration of the overlap structure based on epoxy E51 chains with dynamic disulfide linkages and entanglement of PCL. (**c**) Multi-shape memory circles of welded-multilayered vitrimer composites [141]. Adapted with permission from [141]. Copyright 2024, Elsevier.

**Figure 8 polymers-16-00621-f008:**
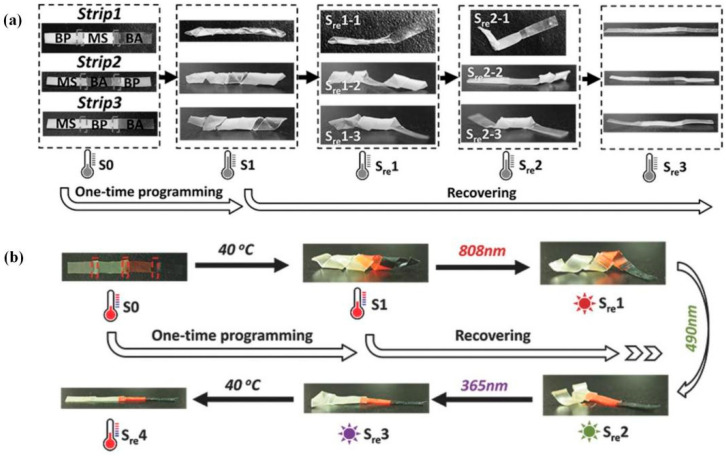
(**a**) Multiple shape changes of strips under different thermal stimuli. (**b**) Multiple shape changes of strips under thermal stimulus and multiwavelength photo-stimulus [148]. Adapted with permission from [148]. Copyright 2024, Wiley.

**Figure 9 polymers-16-00621-f009:**
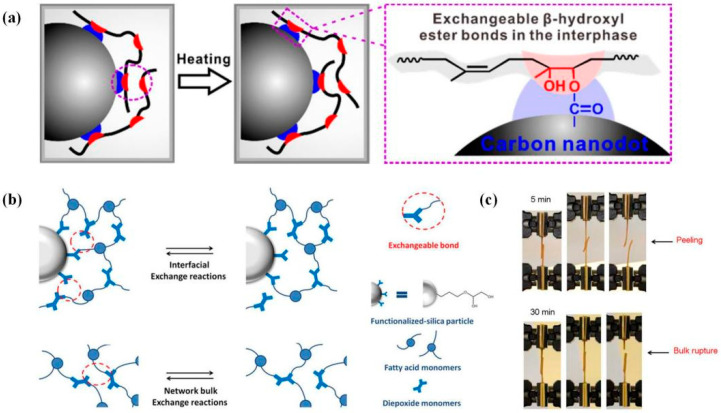
(**a**) Topological rearrangements via exchange reaction in ENR–CD interphase [150]. Adapted with permission from [150]. Copyright 2024, American Chemical Society. (**b**) Adapting surface chemistry of fillers to the epoxy-vitrimer network. Exchange reactions can happen both at the fillers-matrix interface and in the network. (**c**) Lap-shear experiments performed on samples welded at 190 °C for various welding times [82]. Adapted with permission from [82]. Copyright 2024, American Chemical Society.

**Figure 10 polymers-16-00621-f010:**
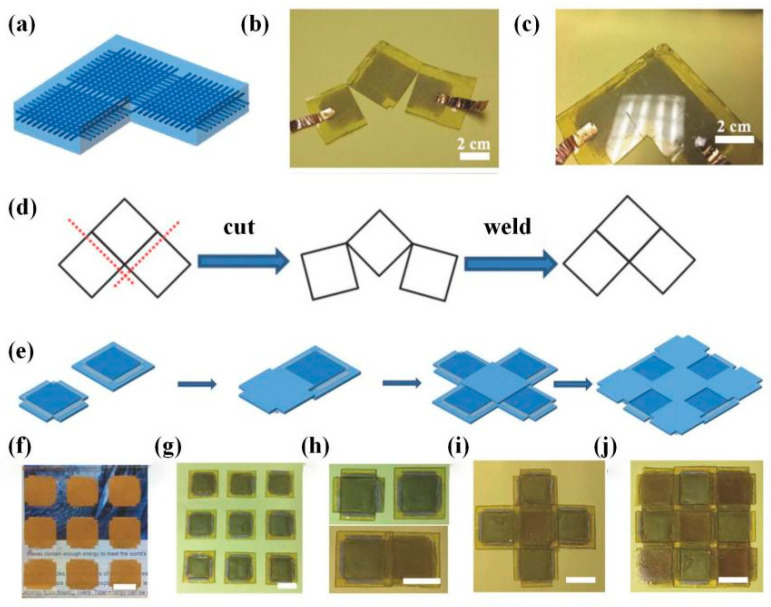
(**a**) Space diagram of the square ruler-like VTENG. (**b**) Strategy for break and recovery. (**c**) Photo of broken VTENG. (**d**) Photo of recovery VTENG. (**e**) Space diagram, and (**f**–**j**) photos of the fabrication method and VTENG assembled with two, five, and nine pieces. Scale bar: 3 cm [110]. Adapted with permission from [110]. Copyright 2024, Wiley.

**Figure 11 polymers-16-00621-f011:**
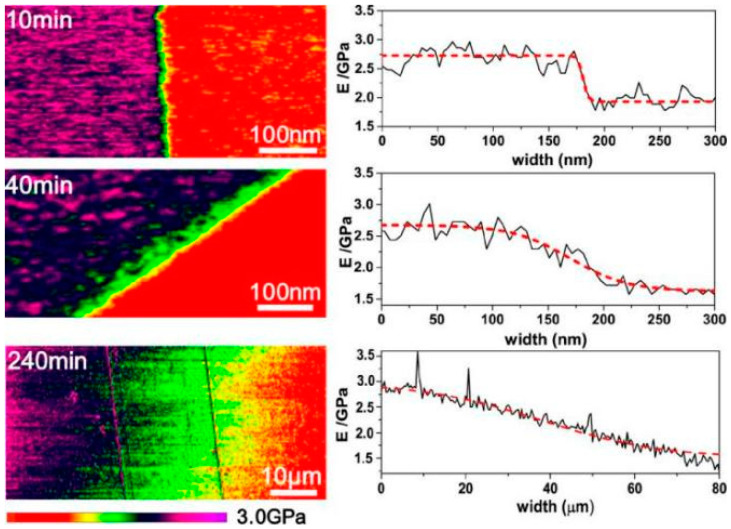
Young’s modulus maps across the interface at 170 °C for different times and the corresponding modulus profiles. The red dash lines hyperbolic tangent function fit the modulus data [155]. Adapted with permission from [155]. Copyright 2024, American Chemical Society.

**Figure 12 polymers-16-00621-f012:**
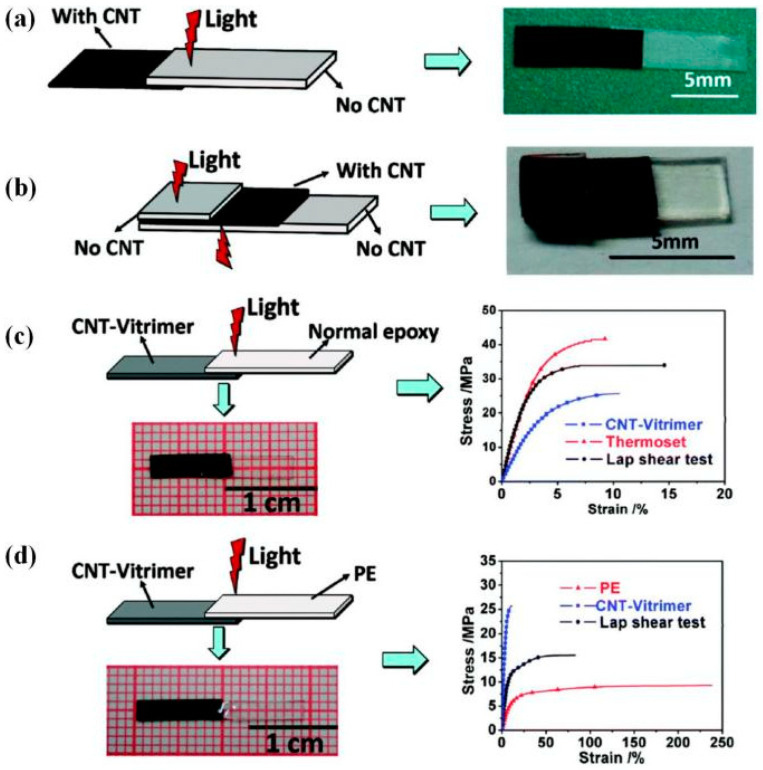
Transmission welding (**a**) joining non-CNT vitrimer with CNT–vitrimer. (**b**) Joining two pieces of non-CNT vitrimer using CNT–vitrimer as an “adhesive”. (**c**) Joining normal epoxy with CNT–vitrimer. (**d**) Joining thermoplastic PE with CNT–vitrimer [107]. Adapted with permission from [107]. Copyright 2024, Royal Society of Chemistry.

**Figure 13 polymers-16-00621-f013:**
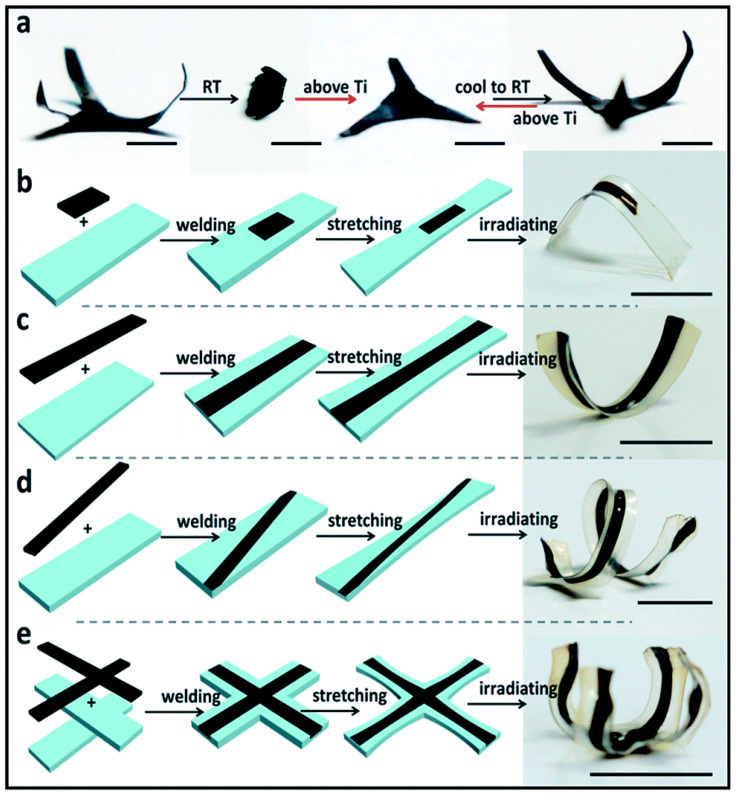
Schematic illustrations of the process used to prepare various welded blank/PDA-xLCE sample films, and optical pictures of the resulting 3D structures. Blank LCE films were welded into different shapes using PDA-xLCE: (**a**) restoration and reversible actuation of deformed dynamic 3D structures of a PDA-xLCE sample. Scale bar: 0.5 cm. (**b**) rectangle in the center, (**c**) line in the middle, (**d**) line across the diagonal, and (**e**) cross-section of two lines on a cross-shaped blank film. Light intensity: 1.0 W/cm^2^. Scale bar: 1 cm [34]. Adapted with permission from [34]. Copyright 2024, Royal Society of Chemistry.

**Figure 14 polymers-16-00621-f014:**
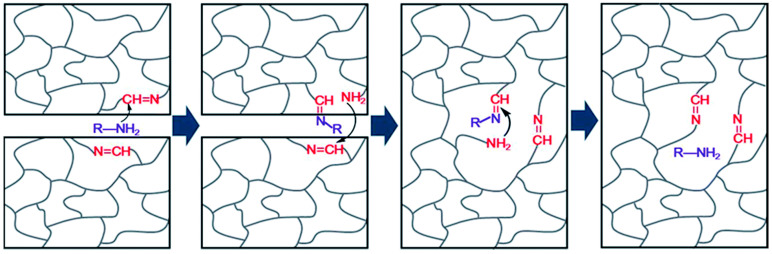
Schematic representation of solvent-assisted welding mechanism via primary amine-promoted imine exchange reaction [31]. Adapted with permission from [31]. Copyright 2024, Royal Society of Chemistry.

**Figure 15 polymers-16-00621-f015:**
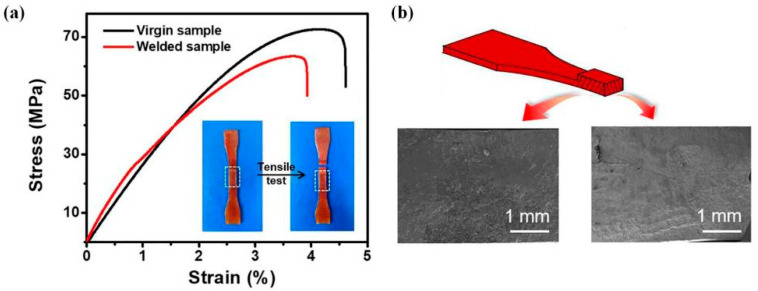
(**a**) Typical stress–strain curves of the virgin sample and welded sample. (**b**) SEM images of the side view and section of the overlapped part after welding [86].

**Figure 16 polymers-16-00621-f016:**
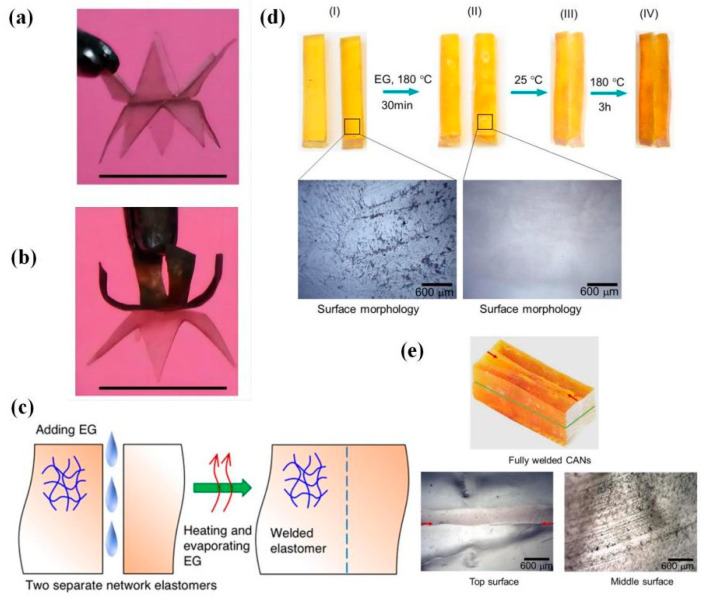
(**a**) Solvent-induced welding of a complicated structure. Scale bar: 1 cm. (**b**) Welding two 3D structures with different components to a complicated structure. Scale bar: 1 cm. [83]. (**c**) Diagram of solvent-assisted welding process for epoxy elastomer. (**d**) Experimental images and microscopic optical images of the fresh-cut and pretreated surfaces. (**I**) Fresh cut samples; (**II**) pretreated surfaces; (**III**) stacking the strips together at room temperature; (**IV**) fully welded strips. (**e**) the microscopic optical images of fully welded CANs: top view and middle view of the interface (red arrows indicate the welded interfaces and green lines represent the middle surface) [84]. Adapted with permission from [31]. Copyright 2024, American Chemical Society.

**Figure 17 polymers-16-00621-f017:**
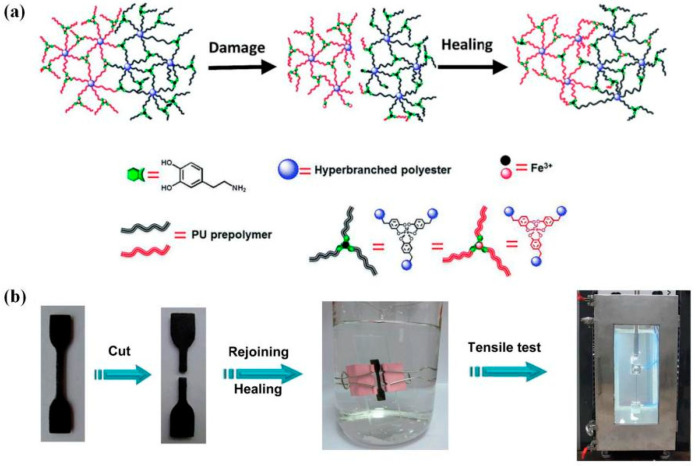
(**a**) Self-weldability of damaged hyperbranched polyurethane DCNs in seawater. Reconnection of the damaged polymer network through the formation of catechol–Fe^3+^ bonds at the interface with the aid of dynamic catechol–iron interactions. (**b**) Dumbbell specimen was cut, recombined, healed, and tested for failure under tension. The entire procedures were conducted in artificial seawater at 25 °C [52]. Adapted with permission from [52]. Copyright 2024, Royal Society of Chemistry.

**Figure 18 polymers-16-00621-f018:**
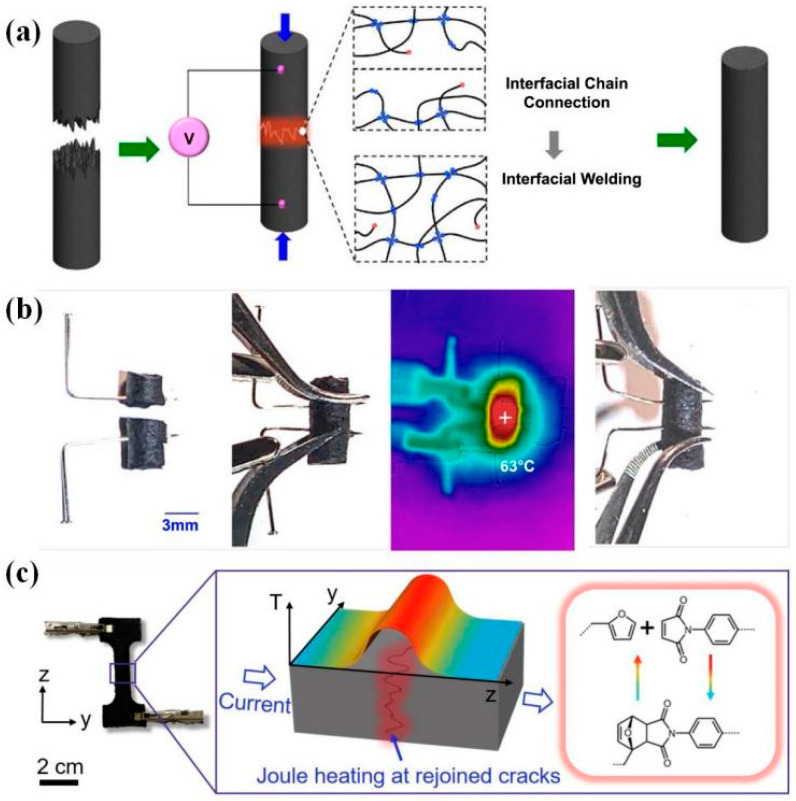
(**a**) Schematic views of the welding process of polyimine composites using Joule heat induced by electricity. (**b**) Appearances of the sample during the welding [165]. (**c**) Schematic illustration of self-healing triggered by localized Joule heating [77]. Adapted with permission from [77]. Copyright 2024, Elsevier.

**Figure 19 polymers-16-00621-f019:**
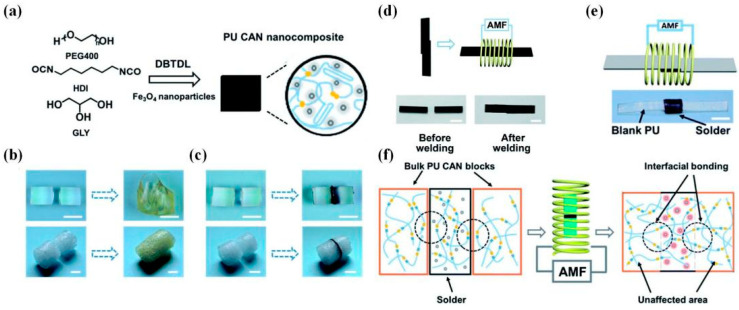
(**a**) Synthesis of the PU CAN nanocomposite. (**b**) Welding of PU CAN bulks and foams via direct heating at 140 °C for 20 min. (**c**) Welding of PU bulks and foams using a magnetothermal responsive solder in the alternating magnetic field. (**d**) Illustration of the magnetothermal effect induced by welding solders. (**e**) A thin layer of solder was sandwiched between two PU films and the films were placed in the alternating magnetic field. (**f**) The mechanism of the magnetothermal effect to weld PU CAN bulks [36]. Adapted with permission from [36]. Copyright 2024, Royal Society of Chemistry.

**Figure 20 polymers-16-00621-f020:**
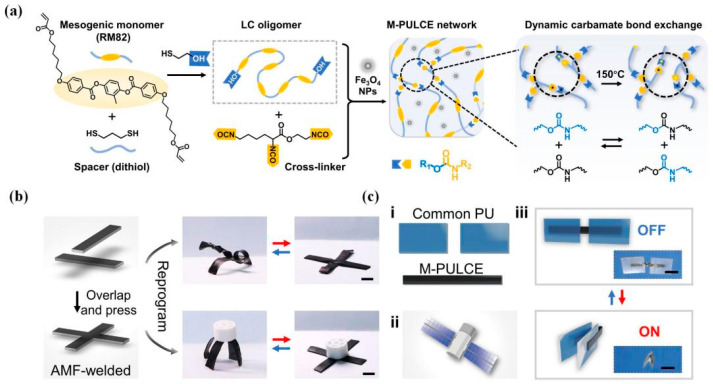
(**a**) Synthesis and mechanism of M-PULCE. (**b**) Integrated M-PULCE actuator with bending/unbending motion, twisting/flattening motion, and coiling/uncoiling motion and its reprogramming. AMF condition: f = 320 kHz, H = 219.27 Gs. (**c**) Multimaterial satellite-like soft actuator. (**i**) Before assembly (two “wings” and the hinge are made of common PU CANs and M-PULCE, respectively), (**ii**) the archetype of satellite, and (**iii**) assembled soft actuators with two “wings” closed when AMF is ON (inset: images). AMF condition: f = 320 kHz, H = 219.27 Gs. Scale bars, 10 mm [11].

**Figure 21 polymers-16-00621-f021:**
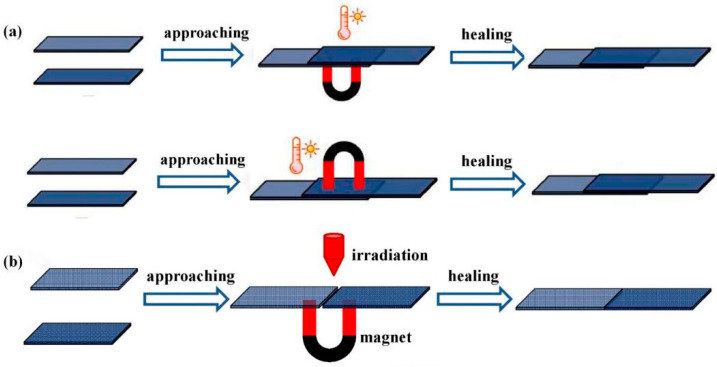
(**a**) Two welding methods under magnet and direct-heating: according to the directions of gravity and magnetic attraction. (**b**) Fracture-welding under magnet and NIR irradiation [38]. Adapted with permission from [38]. Copyright 2024, Elsevier.

**Figure 22 polymers-16-00621-f022:**
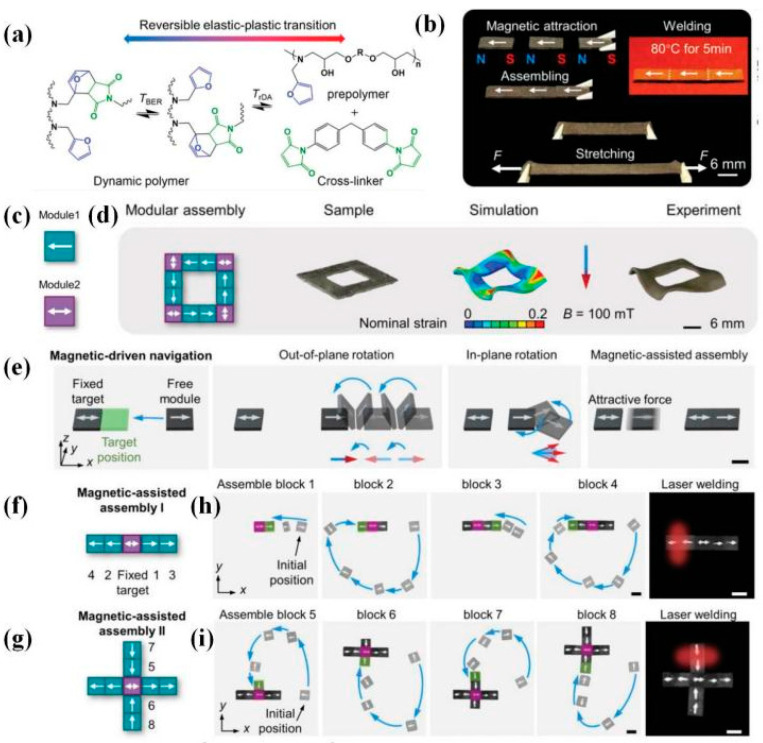
(**a**) Scheme of reversible elastic–plastic transition via network topology transition in the DA-reaction-based DP at different temperatures. (**b**) Images of a long strip assembly consisting of three MDP modules via magnetic attraction followed by infrared (IR) light heating (80 °C for 5 min). (**c**) Schematics of a square single-directional magnetization module and a bidirectional magnetization module. (**d**) Schematic designs, finite-element analysis, and experimental results of various assembled 2D planar structures with programmed magnetization for complex shape morphing of a square annulus structure. (**e**) Mechanism of the magnetic-driven remote navigation and assembling of the MDP modules. (**f**,**g**) Assembling logics of MDP modules for strip shape (**f**) and cross shape (**g**). (**h**) Assembly of the strip structure in (**f**). (**i**) Assembly of the cross structure in (**g**) [37]. Adapted with permission from [37]. Copyright 2024, Wiley.

**Table 1 polymers-16-00621-t001:** Common bond exchange reactions in two topology rearrangement mechanisms.

Dissociative Mechanism	Associative Mechanism
Diels–Alder reaction	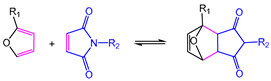	Transesterification	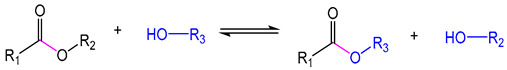
Transcarbomoylation of urethanes	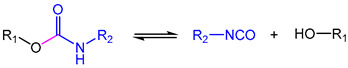	Disulfide Exchange	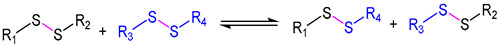
Imine condensation	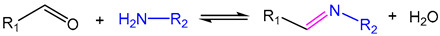	Amine–imine Exchange/Transamination	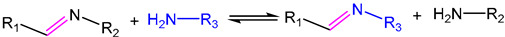
Aminal transamination	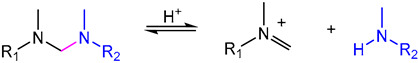	Imine metathesis	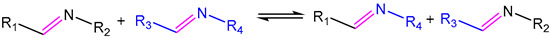
Oxime-promoted transcarbamoylation	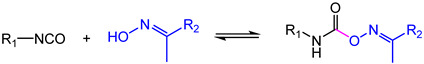	Transcarbonation	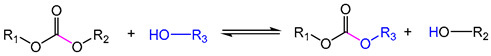
Thioacetalexchange	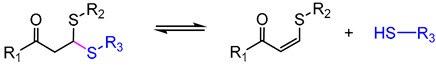	Transamination of vinylogous urethanes and amides	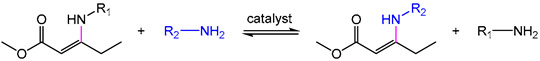
Triazolinedione-indole Alder-Ene	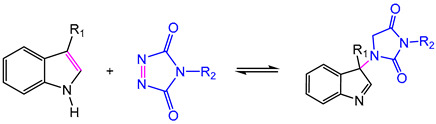	Boronic ester exchange	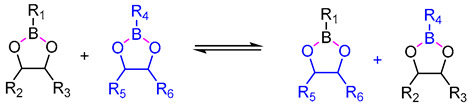
Amine urea exchange	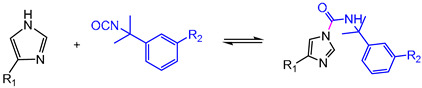	Thiocarbamate exchange	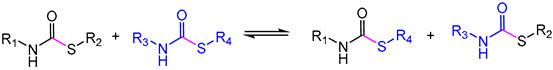
Boronic ester hydrolysis	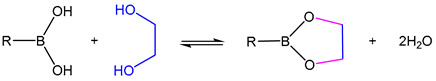	Silyl ether exchange	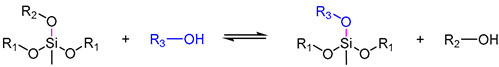

**Table 2 polymers-16-00621-t002:** Overview of weldable DCN systems.

Network Preparation and Property							Weldability Experiment				Ref.
Dynamic Mechanism	Monomer	Cure Agent/ Cross-Linker	Catalyst	Additional Component	T_g_/°C		Process Condition	Sample Size	Virgin Strength /MPa	Welded Strength/ MPa	
					DSC	DMA					
Transesterification	Epoxy E1, E3	Pripol 1040	Zn(acac)_2_	-	11–23	34–38	120 °C 4 h Compressing with the clamp	Overlapped rectangle samples (17.5 mm × 3.0 mm × 1.0 mm), superimposed on a 2.5 mm length.	-	0.4–0.6	[78]
	DGEBA	Pripol 1040	Zn(OAc)_2_	-	-	~15	150 °C 1 h Applying a 25% compression	Overlapped rectangle samples (25 mm × 5 mm × 1.4 mm), superimposed on a 15 mm length.	0.47	0.36	[21]
	DER331	Succinic anhydride	-	-	62.5–72.1	70.3–82.4	150 °C 1 h A hot press	Overlapped fragments from dumbbell films with a thickness of 0.3 mm.	46.5 ± 2.3	47.3	[106]
	Araldite LY564	Aradur 917CH	Zn(acac)_2_·*x*H_2_O	glass fabric	-	-	160 °C 90 min Applying a 500 N or 1320 N load	Overlapped rectangle samples (50 mm × 15 mm × 3 mm), superimposed on a 20 mm length.	-	1.67	[135]
	THFA, AM	Diacrylate prepolymer	TBD	-	2.4–51.1	9.1–40.1	180 °C 2 h A hot press	Overlapped dumbbell samples (75 mm × 5 mm × 2 mm), superposed on a 5 mm length.	14.3–15.0	8.5–11.0	[58]
	HPPA	BPA.GDA	Zn(acac)_2_·*x*H_2_O	-	-	30	180 °C 4 h No pressure	Rectangle samples (30 mm × 5 mm × 1.5 mm) with a hole ~ 5 mm diameter and a circular disc (5 mm × 1.5 mm).	15	14	[28]
	HPPA, DG2A	TMPMP	Miramer A99	-	0~20	-	180 °C 4 h No pressure	Rectangle samples (30 mm × 5 mm × 1.5 mm) with a hole ~ 5 mm diameter and a circular disc (5 mm × 1.5 mm).	4	4	[121]
	HPPA, DG2A	EGMA	Miramer A99	-	-	43	180 °C 4 h	Rectangle samples (30 mm × 10 mm × 1.5 mm) with a hole ~ 5 mm diameter and a circular disc (5 mm × 1.5 mm).	34.1	27.9	[61]
	DER 332	Pripol 1040	Zn(Ac)_2_·2H_2_O	Silica	-	30–37	190 °C 30 min Applying a ~30% strain	Overlapped rectangle samples (30 mm × 5 mm × 1.5 mm), superimposed on a 15 mm length.	-	0.69	[82]
	ESO	GL	TBD	-	8–18	39–64	200 °C 15 min No pressure	Three rectangle pieces making up an “H” pattern.	-	-	[117]
	HPPA	BPA.GDA	Zn(acac)_2_	-	-	1.71–4.91	Dropping a solution to cutting edge and photocuring 180 °C 8 h	Two parts from dumbbell films welded with different types of adhesion interface.	47.4	90°: 43.8	[29]
										45°: 45.1	
	DGEBA	adipic acid	TBD	MWCNTs	Cooling: 39 °C, Heating: 45 °C	~60	IR irradiation (0.84 W/cm^2^, 808 nm) 30 s	Overlapped rectangle samples (10.0 mm × 1.0 mm × 0.08 mm), superimposed on a 2.0 mm length.	25.7	20.5	[107]
	DGEBA	sebacic acid	TBD	CNTs	~50	-	IR irradiation (0.84 W/cm^2^, 808 nm) 60 s	Overlapped rectangle films.	22.1	22.3	[136]
	DGEBA	SA	TBD	ACAT	~40	-	IR irradiation (0.22 W/cm^2^, 808 nm) 30 s	Overlapped films with a thickness of 0.2 mm.	17.9	21.0	[137]
	DGEBA	adipic acid	TBD	-	44	60	Immersing into THF Overlapping two swelling films within glass sheets Evaporating the set for 2 days at room temperature	Overlapped rectangle wet films with an overlap area of 2.0 mm × 2.0 mm.	21.8	20.8	[83]
	Poly(BPA-co-EPI)	HHMPA	-	-	-	-	Immersing into EG solution containing 0.1 wt% TBD at 80 °C for 120 min; Taking out and wiping Evaporating and welding at 100 °C for 80 min under 0.5 MPa pressure	Overlapped rectangle wet films (20.0 mm × 5.0 mm × 1.0 mm), superimposed on a 4.0 mm length.	-	4.61 ± 0.11	[138]
	DGEBA	GA	-	-	-	-	Immersing into THF solution containing 1 wt% TBD at 60 °C for 45 min Taking out and wiping Evaporating and welding with clips in the oven for 60 min	Overlapped rectangle wet films (100.0 mm × 25.0 mm × 2.0 mm), superimposed on a 12.5 mm length.	1.12	1.48	[139]
Boronate Esters	DABo, DAP	PETMP	-	-	-	10–18	65 °C 16 h	Ice cream cone and ice cream	-	-	[85]
Phosphate transesterification	DGEBA	[Bmim]DPPOO	-	-	~70	101.1	190 °C 30 min Applying pressure	Overlapped rectangle samples.	-	-	[140]
Transesterification and disulfide exchange	BGPDS	Sebacic acid	TBD	MWCNTs	43.9–47.9	5.9–16.3	A drop of PETMP NIR irradiation (1.0 W/cm^2^, 808 nm) 30 s	Overlapped rectangle samples.	16.4	16.1	[111]
	DGEBA	DTDA	TBD	Fe_3_O_4_	22–27	25–28	Magnet-based assembling 100 °C 1 h	Overlapped samples, superimposed on a 7.5 mm length.	-	~8 N	[38]
							Magnet-based assembling NIR irradiation (5 mW, 808 nm) several seconds	Samples with their cross-sections in contact.	1.4	1.0	
Disulfide exchange	Epon resin 828	EDDET, Polysulfide oligomer	TBD	-	-	−0.7	95 °C 30 min	Two rectangle samples (30 mm × 5 mm × 1.5 mm), with their cross-sections in contact.	1.28	1.23	[110]
	ESO	APD	-	-	2.9–12.1	26.2–33.7	120 °C 30 min	Dumbbell samples with a width of 4.0 mm and a thickness of 0.5 mm, superimposed on a 5 mm length.	3.2	3.3	[112]
	E51	dithioaniline	-	PCL	~21	~49	130 °C 1 h	Overlapped rectangle samples.	-	-	[141]
	DEGBA	Oleylamine, 4-AFD	-	CNT	66	73	NIR irradiation 1 min	Rectangle samples with their cross-sections in contact.	2.6	2.2	[113]
	EPS25	EDDET, PETMP	DMAP	-	-	−35	A static magnetic field 100 °C 900 s	Broken dumbbell parts brought into contact.	0.12	0.09	[108]
Disulfide exchange and hydrogen bonding	HEA, PEG-1000, HEDS	IPDI	-	-	-	-	80 °C 12 h	Cutting a dumbbell sample into two parts.	3.39 ± 0.09	3.22 ± 0.40	[125]
	HEDS, IPDI, PEG400	TEA	-	-	29.8	-	Sunlight from 10:00 a.m. to 4:00 p.m. in July	Overlapped fragments from dumbbell films.	9.6	9.2	[60]
Disulfide exchange and imine exchange	PB, AET	SCHO	-	MWCNTs	−24.73	27.6	100 °C 1–2 h applying a 200 g load	Overlapped rectangle samples (30.0 mm × 6.0 mm × 0.7 mm), superimposed on a 10 mm length.	-	2.7–4.2	[26]
	DSEP	DDM, D400	-	-	-	129	160 °C 20 min	Cutting a sample in two parts and then overlapping.	55.0	53.5	[119]
Imine exchange	PB, AET, DMPA	Vanillin derivatives	-	-	14.73	39.65	100 °C Applying a 200 g load	Overlapped rectangle samples (30.0 mm × 6.0 mm × 0.55 mm), superimposed on a 10 mm length.	14.5	12.2	[27]
	DGEBA, vanillin, aminophenol	Jeffamine	-	-	-	71	Preheated at 100 °C for 60 s under 10 N force 120 °C for 4 h	Overlapped rectangle samples (50.0 mm × 12.5 mm × 0.4 mm), superimposed on a 3.2 mm length.	45.8	46.5	[88]
	DIDG, TMPTE	Jeffamines (Jeff230, JeffD400, and JeffT403)	-	-	66.8–92.8	-	180 °C 2 h 3 MPa	Overlapped rectangle samples.	40.0	27.9	[30]
	DGEBA	Jeffamine D230, TA	-	-	52	54	150 °C 24 h	Splitting the sample into two parts with their cross-sections in contact.	51.4	34.7	[142]
	DER 331	Jeffamine EDR-148, TPA	-	carbon fiber	57	63	180 °C 1 h 1 MPa	Overlapped rectangle samples.	-	8.33	[143]
	TPA, MXDA	TREN	-	ACAT	-	107	110 °C 30 min Applying a 200 g load	Overlapped rectangle samples.	53.5	51.4	[144]
							0.1 mL EDA to overlapping region 110 °C 12 min Applying a 200 g load	Overlapped rectangle samples.		51.9	
							0.1 mL EDA to overlapping region NIR light (808 nm, 1.5 W/cm^2^) 30 s	Overlapped rectangle samples.	54.9	53.0	
	DGEBF	TAD, AB	-	-	-	100.2	One drop of DETA solution in DMF solvent (10 mg/mL) to overlapping region 140 °C 1 h	Overlapped fragments from dumbbell films, superposed on a 10 mm length.	72.8	63.9	[86]
	6FDA, ODA	DETA	-	-	-	217	One drop of DETA solution in DMF/ acetonitrile solvent (*v*/*v* = 1:1, 10 mg/mL) to overlapping region Heat pressing using a temperature program (50 °C, 60 °C, 80 °C, 2 h for each temperature)	Overlapped dumbbell samples (Width × Thickness = 80 mm × 5 mm), superimposed on a 10 mm length.	82.7 ± 2.8	78.2 ± 0.7	[31]
	ODPA, DPD	TREN, DETA	-	-	-	137	One drop of DETA solution in DMF/ acetonitrile solvent (*v*/*v* = 1:1, 10 mg/mL) to the contact area Heat pressing using a temperature program (50 °C, 60 °C, 80 °C, 2 h for each temperature)	Dumbbell films were cut into two pieces, which were then put together in contact with a crack width of ~400 mm.	69.37	68.6	[32]
Diels–Alder reaction	NGDE, FA	BMI	-	b-CNTs	8	-	DC power (24 V or 16 V) 10 min	Rectangle samples (20.0 mm × 5.0 mm × 0.6 mm).	7.2	6.9	[77]
									5.9	5.6	
	PEGDGE, FA	Commercial bismaleimide cross-linker	-	NdFeB	−35	−20	Magnetic attraction (100 mT) Infrared (IR) light heat (80 °C) 20 min	Rectangular samples (25 mm × 4 mm × 0.9 mm).	0.16	0.15	[37]
Hindered urea bond	IBOA, PEA	PPIA	-	-	−10–70	3–8	160 °C 2 h	Kresling-patterned material modules.	-	-	[46]
Piperidine-urea bond exchange	IPDI, PIP	Castor oil	-	-	-	30.1	150 °C 30 min	Overlapped dumbbell samples (Width × Thickness = 4.0 mm × 0.5 mm), superimposed on a 5 mm length.	16.4	14.3	[23]
Thiourea bond exchange and hydrogen bonding	TUEG	TGIC	TBD	-	27.8	46.4	160 °C 2 h Applying a ~10% compression	Overlapped area of 5 mm× 5 mm, with a thickness of 0.4 mm.	23 ± 1	19 ± 0	[59]
Oxime–carbamate exchange	BQDO	THDI, Jeffamine D230	-	-	-	T_g1_: 65 °C, T_g2_: 110 °C	NIR irradiation (2.7 W/cm^2^, 808 nm) 150 s	Overlapped rectangle samples (20.0 mm × 4.0 mm × 0.6 mm), superimposed on a 4.0 mm length.	-	1.2	[145]
Thiocarbamate bond exchange	MMA	synthesized by DIPEA-catalyzed click reaction of EDDET and IEM	-	-	83	-	150 °C 4 h 6 MPa	Two half dumbbell samples with their cut cross-sections in contact.	61.3	50.3	[103]
Transcarbamoylation	HDI, PEG400	GLY	DBTBL	Fe_3_O_4_	−15	-	AMF (f = 495 kHz, H = 127.331 Gs) 120 s	A solder placed between the two PU bulks.	3.4	3.1	[36]
	Hydroxyl-terminated LC-oligomer	LTI	DBTBL	Fe_3_O_4_	−8.8	-	AMF 5 min	Overlapped rectangle samples (Width × Thickness = 2.25 mm × 0.40 mm).	28.9	22.8	[11]
Hydrogen bonding	DGEBA, TGMDA	Pripol 1040, UDETA	2-MI	-	11–16	~23	22 °C 24 h	Dog-bone specimens.	3.7	2.0	[57]
Hydrogen bonding and ionic bonding	UMA, AA	ZDMA	CQ, EDMAB	-	10–58	20–83	90 °C 12 h	-	-	-	[22]
Host–guest interaction	BADA	AAAB	-	β-CD- Graphene Nanosheets	-	88.9	120 °C 30 min In the dark	Two damaged pieces.	13.5	11.5	[54]
							NIR (2 W) 10 min			8.6	

## Data Availability

Data is contained within the article.

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
