# Peer review of "External Stimuli-Induced Welding of Dynamic Cross-Linked Polymer Networks"

_polymers, 2024, doi:10.3390/polym16050621_

Round 1
Reviewer 1 Report
Comments and Suggestions for Authors
In this review, Liu et al. reported different strategies for external stimuli-induced welding for mainly epoxy and acrylate-based dynamic networks. These strategies rely on thermal, photo, solvent and electrical/magnetic induced welding of these type of polymers. They encompass different dynamic bonds (esters, disulfide, imines, PU,…) that have been proved to be suitable for welding. They extensively explain the work that has been done in the academic world providing adequate and current references for each case. The manuscript is well-written, and very understandable and its extent is appropriate. Therefore, I recommend the acceptance of this review for the Polymers Journal after some minor revisions:
1) The authors entitle this review referring to “cross-linked polymer networks”. This may induce to thermosetting materials. Moreover, they explain the strategies used in DCN. For this reason, I recommend the introduce the word “dynamic” or DCN somewhere in the title.
2) In page number 4, they refer to disulfide exchange as an associative-type of CAN. Nevertheless, there is still a current debate about the mechanism of this exchange reaction. Probably depending on the environment or the type of disulfide the exchange. Moreover, some reports in the literature demonstrate that this type of CANs can be vitrimer-like. In addition, the authors mostly refer to aromatic disulfide metathesis. Nevertheless, some work regarding potential welding abilities has been done in epoxy networks containing aliphatic disulfide bonds.
3) In Table 1, the transesterification scheme, needs the presence of an alcohol. In the same table, the transamination reaction should be specified for which group. The word transamination alone can be misleading with other type of exchange mechanism like amine-imine exchange.
4) In the introductory part the authors should explain more in detail the relationship between CANs and welding ability as well as in which fields the welding can be potentially used. Currently it seems an introduction of covalent adaptable networks. The reviewer do not also understand which is the point of the explanation about the Tv in vitrimers in this review.
5) In page number 5, the authors refer the work done by Leibler and co-workers (reference 5) by mentioning the use of DGEBA and glutaric anhydride. However, this is not correct. In this work they did not used anhydrides but different carboxylic acids.
6) In page number 6, the authors state that they will “focus on the most common material systems based on different radical initiation ways of network polymerization”. Nevertheless, they explain different epoxy networks that has been polymerized through an anionic and cationic methods. This sentence should be modified.
7) In page number 15, second paragraph, there is a Fig. without the number.
8) Some Figures are not cited in the text (i.e., Figure 18, 20 and 21). Citations should be added prior to the corresponding Figures.
9) In the summary and outlook the authors state that the academy should move towards fabricating DCN in a more sustainable and environmentally friendly way. Nevertheless, they subsequently mention the use of benign catalysts. The reviewer consider that this state should be modified since the best environmental option is to synthesize CANs without the use of external catalysts since it can induce to exuding, boiling or dripping. Moreover, the use of sustainable and non-toxic reagents for the synthesis of these materials should be mentioned.
Reviewer 2 Report
Comments and Suggestions for Authors
The review presents recent advances in welding processes of dynamic cross-linked networks (DCNs) under external stimuli, including innovative covalent adaptable networks (CANs) that bridge the gap between thermosets and thermoplastics. This review is well organized and describes the welding of epoxy DCNs and acrylate DCNs, including some biobased CANs, such as epoxidized soybean oil. Biobased materials and welding processes are relevant nowadays to the environment and human health. However, there are still some issues that need further clarification. Please consider the following observations:
1. In my opinion, more biobased materials should be included in the review as welding of biobased CANs is a combination of green engineering and green chemistry concepts to reduce the negative impact on the environment and human health caused by the chemical industry.
2. Self-weldable CANs with covalent bonds without external stimuli need to be more emphasized in Section 2.2.
3. The abbreviations section should be reviewed as there are some correctional errors, such as uppercase and lowercase letters, and some other errors, e.g. “GLY: giycerine”. Should it be glycerine?
4. Conclusions need to be reorganized by highlighting at least three important ones expressing the future research more clearly.
Comments on the Quality of English LanguageThe English language is fine. However, some writing errors should be considered.
Round 2
Reviewer 1 Report
Comments and Suggestions for Authors
The authors have answered the questions and comments properly and in a very understandable way. Currently, I consider that the manuscript has a higher quality and its easily readable, thus I recommend the acceptance of this manuscript in the presence form in the Polymers Journal.
Reviewer 2 Report
Comments and Suggestions for Authors
I have no questions. Thanks for authors for the response.